# AetherCode: Evaluating LLMs' Ability to Win In Premier Programming Competitions

**Zihan Wang**[1,2]**, Jiaze Chen**[3]**, Zhicheng Liu**[3]**, Haojie Pan**[3]**, Markus Mak**[3]**, Yidi Du**[3]**,
Geonsik Moon**[3]**, Aaron Tua**[3]**, Kunshuo Peng**[3]**, Jiayi Lu**[3]**, Boqian Zou**[3]**, Chenyang Ran**[3]**,
Guang Tian**[3]**, Shoutai Zhu**[3]**, Yeheng Duan**[3]**, Zhenghui Kang**[3]**, Zhenxing Lin**[3]**, Shangshu Li**[3]**,
Qiang Luo**[3]**, Qingshen Long**[3]**, Zhiyong Chen**[3]**, Yihan Xiao**[3]**, Yurong Wu**[3]**, Daoguang Zan**[3]**,
Hongyan Li**[1,2]**, Mingxuan Wang**[3]**, Ming Ding**[3]

[1]State Key Laboratory of General Artificial Intelligence, Peking University

[2]School of Intelligence Science and Technology, Peking University

[3]ByteDance

zh.wang@stu.pku.edu.cn

## Abstract

Competitive programming has emerged as a critical benchmark for evaluating the reasoning and coding capabilities of Large Language Models (LLMs). Despite impressive progress on existing benchmarks, we argue that current evaluations overstate model proficiency, masking a substantial gap between LLMs and elite human programmers. This gap arises from two key limitations: insufficient difficulty and scope of benchmark problems, and evaluation bias from low-quality test cases. To address these shortcomings, we present **AetherCode**, a new benchmark that draws problems from premier programming competitions such as IOI and ICPC, offering broader coverage and higher difficulty. AetherCode further incorporates comprehensive, expert-validated test suites built through a hybrid of automated generation and human curation, ensuring rigorous and reliable assessment. By combining challenging problem design with robust evaluation, AetherCode provides a more faithful measure of LLM capabilities and sets a new standard for future research in code reasoning.

## 1 Introduction

Competitive programming is widely regarded as a crucial benchmark for evaluating the reasoning and coding capabilities of Large Language Models (LLMs) (OpenAI et al., 2025). Solving complex competitive programming problems demands not only sophisticated reasoning abilities but also knowledge from diverse domains, including mathematics, data structures, and algorithms. Recent years have witnessed rapid advancements in the reasoning capabilities of LLMs, a key indicator of which is their success on a majority of existing code reasoning benchmarks. State-of-the-art models now achieve over 90% *Pass@1* accuracy on MBPP (Austin et al., 2021) and HumanEval (Chen et al., 2021), and over 80% on LiveCodeBench (Jain et al., 2025). These encouraging developments might lead one to ask: has competitive programming been mastered by LLMs?

In this paper, we argue that a significant gap still exists between the performance of LLMs and top-tier human competitors in programming contests. We propose that the perception of LLM dominance stems primarily from the limitations in the breadth and rigor of current code reasoning benchmarks, which are no longer sufficient to fully assess the capabilities of today's increasingly powerful models. Specifically, we identify two main shortcomings in existing benchmarks:

- **Insufficient Difficulty and Scope.** Early benchmarks such as HumanEval (Chen et al., 2021) and MBPP (Austin et al., 2021) consist of basic coding tasks, for instance, sorting or reversing a list, which present minimal reasoning challenges for state-of-the-art LLMs. More recent "competition-level" benchmarks often source problems from a limited set of websites. For example, LiveCodeBench (Jain et al., 2025) collects problems mainly from LeetCode and AtCoder, while CodeELO (Quan et al., 2025) and LiveCodeBench Pro

(Zheng et al., 2025) originate solely from CodeForces. The problems from these websites have inherent limitations. LeetCode problems are generally easier and often require only the implementation of a single function rather than a complete program. CodeForces contests, which typically feature 5-7 problems within a 2-3 hour timeframe, constrain the design space for problem setters, for example, leading to a scarcity of problems that require complex, large-scale implementations.

- **Evaluation Bias from Low-Quality Test Cases.** Inaccurate verifiers introduce bias into the evaluation (Vendrow et al., 2025). The correctness of a piece of code is verified using a comprehensive set of test cases (input-output pairs). An incomplete test suite may fail to detect incorrect submissions, particularly those with subtle flaws, such as the mishandling of corner cases or solutions that exceed time limits under specific, extreme conditions. Consequently, designing high-quality test cases is a huge challenge that requires a deep understanding of potential failure points, a skill typically honed through extensive competitive programming experience. Most past benchmarks lack sufficiently rigorous test cases. HumanEval (Chen et al., 2021) and MBPP (Austin et al., 2021), for instance, rely on a small number of handwritten test cases. Others, including EvalPlus (Liu et al., 2023), CodeContests (Li et al., 2022), and LiveCodeBench (Jain et al., 2025), employ naive test case generation pipelines, such as random mutation, which fall far short of the quality of expert-designed test suites. Furthermore, recent research (Wang et al., 2025b) has revealed issues with test case correctness itself; for example, many test cases in the Code-Contests dataset do not adhere to the problem's constraints, causing even correct solutions to fail. It is worth noting that some recent benchmarks, such as CodeELO (Quan et al., 2025) and LiveCodeBench Pro (Zheng et al., 2025), have attempted to leverage the official CodeForces judging service to indirectly access its high-quality, expert-crafted test cases. However, this approach presents two significant issues. First, it raises compliance risks, as CodeForces explicitly prohibits the use of crawlers on its judging interface. Second, this method is constrained by submission frequency limits, which impedes agile and flexible experimentation. Therefore, we contend that an open-source benchmark with high-quality, self-contained test cases remains critically important for the LLM community.

To address these challenges, we introduce AetherCode, a new benchmark with the following key contributions:

**Problem Curation from Top-Tier Competitions.** AetherCode is the first benchmark to systematically collect latest problems from premier programming competitions worldwide, including the Olympiad in Informatics (OI) and the International Collegiate Programming Contest (ICPC). Our process involved a comprehensive collection, meticulous cleaning, and format conversion of problems from PDF to a Markdown+LaTeX structure. Each problem statement was manually proofread for correctness, and a team of competitive programming experts annotated each problem with classification tags.

**High-Quality Test Case Generation.** We developed a hybrid methodology, combining automated generation with expert annotation, to create high-quality test cases for every problem. We evaluated the correctness and comprehensiveness of our test cases by validating them against a large corpus of collected solutions, enforcing a standard of zero false positives and zero false negatives.

This paper is organized as follows: Section 2 details the benchmark curation process. Section 3 presents our evaluation results. Section 4 presents some related work, and Section 5 concludes the paper with comments for future research.

## 2 BENCHMARK CURATION

This Section details the curation process of the AetherCode Benchmark. The overall curation process is illustrated in Fig. 1. Then, Sections 2.1 and 2.2 describe the specifics of problem collection and categorizing, respectively. Section 2.3 explains how we construct high-quality test cases for each problem.

Table 1: Comparison between AetherCode and other code reasoning benchmarks

| Dataset | Difficulty | # Problems | Updates | Test Cases Construction | Source |
|---|---|---|---|---|---|
| HumanEval (Chen et al., 2021) | ★ | 164 | ✗ | Handcrafted | Original |
| MBPP (Austin et al., 2021) | ★ | 974 | ✗ | Handcrafted | Original |
| APPS (Hendrycks et al., 2021) | ★★★ | 5,000 | ✗ | Crawled | CodeForces, AtCoder *etc.* |
| USACO (Shi et al., 2024) | ★★★ | 307 | ✗ | Publicly accessible | USACO |
| CodeContests (Li et al., 2022) | ★★★ | 165 | ✗ | Mutation | CodeForces, AtCoder *etc.* |
| LiveCodeBench (Jain et al., 2025) | ★★ | 1055 | ✔ | Semi-automatic | LeetCode, AtCoder |
| CodeELO (Quan et al., 2025) | ★★★ | 387 | ✔ | - | CodeForces |
| LiveCodeBench Pro (Zheng et al., 2025) | ★★★ | 584 | ✔ | - | CodeForces |
| AetherCode | ★★★ | 456 | ✔ | G-V Agent & Experts | Premier Contests |

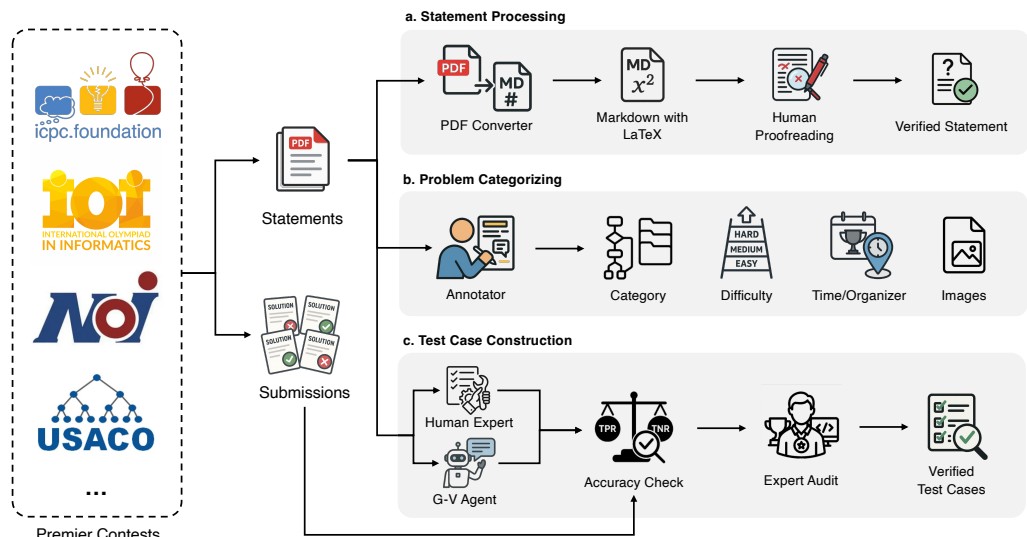

Figure 1: **Curation process of AetherCode.** (a) We begin by converting the collected problem statements from PDF to a Markdown+LaTeX format, which is then manually proofread for accuracy. (b) Each problem is then categorized by its algorithm type and difficulty level, and we also compile additional metadata, such as the time and the organizer of the contest. (c) To ensure quality, we use a G-V Agent in conjunction with human experts to annotate test cases. The accuracy of these test cases is then evaluated against a collected set of solutions and further audited by gold medalists and professional problem setters.

## 2.1 PROBLEM COLLECTION

We source our problems from premier programming competitions worldwide rather than from online programming websites. Based on their target audience, these competitions can be broadly categorized into two main series: the Olympiad in Informatics (OI) series, which is aimed at pre-college school students, and the International Collegiate Programming Contest (ICPC) series, which is designed for college students.

**OI Series.** The Olympiad in Informatics is a series of competitions aimed at popularizing computer science knowledge among high-school students and cultivating outstanding talents in computer science. The OI competitions usually require participants to solve algorithm-related problems by programming. Take the International Olympiad in Informatics (IOI), the top-level event of OI, as an example. Each contestant competes individually, and each country can send up to 4 players. During the two-day competition, players need to independently solve 3 problems within 5 hours each day, mainly using C++. Furthermore, various countries and regions host their own national or regional OI competitions, such as the National Olympiad in Informatics (NOI) in China and the USA Com-

puting Olympiad (USACO) in the United States. Top-performing contestants in these competitions earn the opportunity to advance to the IOI.

**ICPC Series.** The ICPC is the oldest, largest, and most prestigious university-level programming contest in the world. Each team consists of up to 3 students and uses one computer to solve 10 - 13 problems in 5 hours, using programming languages such as C, C++, Java, or Python. The team that correctly solves the most problems with the least total time wins.

The world is divided into several regions for the ICPC. In Europe, there are Central Europe (CERC), North Europe (NWERC), South-East Europe (SEERC), and South-West Europe (SWERC) regions. Other regions include Asia-Pacific, Asia East Continent, North America, Latin America, Africa, and Arab region, etc. The ICPC is a multi-tiered event. First, there are **regional contests** held worldwide from September to November each year. The top-performing teams in the regional contests advance to the **regional finals or championships**. Then, the best teams from these finals or championships qualify for the ICPC **World Finals**, which is usually held from April to June each year. This is the highest-level stage of the ICPC, where the best teams from around the world compete for the championship.

In addition to the official ICPC events, we also incorporated problems from other large-scale and renowned collegiate programming contests, such as the China Collegiate Programming Contest (CCPC).

For each problem, we collected the following components:

- **Problem Statement.** The statement typically comprises a title, a detailed problem description, input/output specifications, sample inputs and outputs with explanations, data range constraints, and time/memory limits. The majority of the problem statements was originally in PDF format. To enhance comprehension for LLMs, we converted these PDFs into a Markdown format with LaTeX for mathematical notations. Each converted file was then manually proofread to ensure its accuracy.

- **Solutions.** We curated a collection of over 30,000 human-written solutions for these problems, encompassing both correct and incorrect submissions. For each problem, we ensured a minimum of 5 correct and 20 incorrect solutions. The primary purpose of collecting these solutions is to evaluate the quality of the subsequently generated test cases, a process detailed in Section 2.3.

- **Test Cases.** A minority of the competitions, e.g., USACO, publicly released their official test cases, which we collected and standardized. For problems where official test cases were not available, we constructed our high-quality test cases. The methodology for this construction is described in Section 2.3.

- **Metadata.** We also gathered auxiliary information, such as the date of the competition (for decontamination purposes) and human contestant performance data (to facilitate difficulty assessment), among other available data points.

The data charactistics and statistics of AetherCode v1 are presented in Table 2.

## 2.2 PROBLEM CATEGORIZATION

Beyond curating problems, an equally critical step in constructing AetherCode was the systematic categorization of each problem to ensure comprehensive coverage and facilitate fine-grained evaluation. To this end, we adopted a multi-dimensional categorization framework designed with the input of competitive programming experts:

**Difficulty Segmentation.** Problems were divided into four levels of difficulty: *Easy*, *Medium*, *Hard*, and *Extreme*. This classification was guided by expert judgment as well as official contest results. Most of the problems are distributed roughly evenly among *Easy*, *Medium*, and *Hard*. Notably, problems that no human contestant was able to solve during a competition were specially classified as *Extreme*, representing challenges that push the boundaries of algorithmic reasoning. The number of problems under each difficulty level is presented in Fig. 2. This difficulty classification is judged entirely from the perspective of humans rather than being classified by LLM's performance. This is because we want to provide a perspective to study how the difficulty for LLMs differs from the

difficulty in the eyes of humans. Specifically, we rank problems within the same contest based on the number of participants who successfully solved them. For contests without leaderboards, as well as for determining the relative difficulty order across different contests, we rely on expert evaluation. Finally, based on the overall difficulty ranking of all problems, we divide the dataset into three roughly equal categories: *Easy*, *Medium*, and *Hard*.

**Temporal and Contextual Dimensions.** Each problem was annotated with metadata to enable both decontamination and longitudinal analysis of model performance: (1) **Date of the contest**, allowing chronological tracking of trends in problem design and model capabilities. (2) **Organizer and competition type**, primarily distinguishing between Olympiad in Informatics (OI) and International Collegiate Programming Contest (ICPC) series. (3) **Competition scope**, categorizing contests as regional-level, national-level, or worldwide.

**Problem Properties.** Some problems require additional considerations beyond a standard input–output interface: (1) Problems dependent on visual or image-based input were excluded from the benchmark. (2) Problems requiring special judges (custom checkers) were explicitly labeled to ensure proper handling during evaluation.

**Algorithmic and Domain Categories.** To capture the breadth of algorithmic knowledge tested in programming contests, we implemented a hierarchical taxonomy as shown in Appendix B. The first level consists of ten categories: Algorithm Basics, Search, Dynamic Programming, Strings, Mathematics, Data Structures, Graph Theory, Computational Geometry, Common Techniques, and Problems on Trees. The number of problems corresponding to these ten major categories is shown in the Figure 2. The second level has 144 categories, which are presented in Appendix B. It includes more detailed algorithm tags. For example, the major category "Mathematics" contains several sub-categories such as Number Theory, Linear Algebra, Probability, Game Theory, Combinatorics, and Polynomials. Problems can belong to multiple categories to reflect their cross-disciplinary nature.

This structured categorization enables targeted evaluation of model strengths and weaknesses while also ensuring that AetherCode serves as a scalable resource for future research. In particular, it allows progress to be tracked across difficulty levels, problem types, and algorithmic domains, providing a more comprehensive understanding of model capabilities.

## 2.3 TEST CASE CONSTRUCTION

Recent studies (Liu et al., 2023; Wang et al., 2025b) have highlighted concerns regarding the quality of test cases in several existing code datasets. For instance, benchmarks such as MBPP (Austin et al., 2021) and HumanEval (Chen et al., 2021) include only a limited number of handwritten test cases per problem. Others, like CodeContests (Li et al., 2022) and EvalPlus (Liu et al., 2023), rely on naive methods such as mutation to generate test cases. Consequently, such test cases are insufficient for comprehensively evaluating the correctness and efficiency of a program. Therefore, we contend that the quality of test case construction is a critical factor determining the overall quality of a benchmark.

Notably, some recent benchmarks (Quan et al., 2025; Zheng et al., 2025) directly utilize the Code-Forces's judging service for evaluation. This approach allows them to indirectly access high-quality test cases created by professional problem setters, thereby circumventing the challenge of test case construction. However, this method presents potential compliance risks, as CodeForces explicitly prohibits the use of crawlers on its judging interface. Furthermore, this approach is constrained by submission frequency limits, which impedes agile and flexible evaluation. Therefore, we argue that a benchmark equipped with its own high-quality test cases remains critically important for the LLM community.

To ensure AetherCode possesses sufficiently high-quality test cases, we approached the task from two perspectives. First, we established more stringent evaluation criteria for test case quality, which is presented in Section 2.3.1. Second, we employed a hybrid approach, combining automated generation with expert annotation, to construct the test cases, which are presented in Sections 2.3.2 and 2.3.3. A deteiled procedure of test case generation is presented in Appendix C.

### 2.3.1 Test Case Quality Assessment

Previous research on test case quality has predominantly focused on quantity, operating under the assumption that a greater number of test cases correlates with higher quality (Li et al., 2022; 2023). However, recent studies (Wang et al., 2025b) indicate that quantity is not a direct proxy for quality. This discrepancy arises from two primary issues. First, test cases in some older datasets, despite their volume, suffer from significant correctness issues, often violating the problem's explicit constraints. Second, conventional test case generation methods that merely amass large volumes of random data fail to provide adequate coverage of various special and corner cases.

Consequently, we depart from evaluating test cases by their quantity and instead propose a direct assessment of their ability to discriminate between correct and incorrect solutions. In our framework, we conceptualize the entire test suite for a problem as a binary classifier, that is, a classifier that distinguishes between correct and incorrect solutions. We then evaluate the performance of this classifier using a large, curated collection of both correct and incorrect submissions. We adopt the True Positive Rate (TPR) and True Negative Rate (TNR) as our primary evaluation metrics.

$$\text{TPR} = \frac{\text{True Positive}}{\text{True Positive} + \text{False Negative}} = \frac{\text{Number of Passed Correct Solutions}}{\text{Number of Correct Solutions}} \quad (1)$$

$$\text{TNR} = \frac{\text{True Negative}}{\text{True Negative} + \text{False Positive}} = \frac{\text{Number of Rejected Incorrect Solutions}}{\text{Number of Incorrect Solutions}} \quad (2)$$

The TPR measures the **correctness** of the test cases; a high TPR indicates that correct solutions are not erroneously failed, which is expected when the test cases themselves are valid. Conversely, the TNR measures the **comprehensiveness** or **coverage** of the test cases, quantifying their ability to detect (or "hack") incorrect solutions.

By employing a hybrid approach that combines automated generation with expert curation, we have achieved a 100% TPR and 100% TNR on our collected solution set. This signifies that all collected correct solutions pass our test cases, while all collected incorrect solutions are successfully rejected. To the best of our knowledge, AetherCode is the first benchmark that sets such a high standard for test cases.

### 2.3.2 Automatic Construction of Test Cases

We employed the Generator-Validator (G-V) Agent System (Wang et al., 2025b) to automatically construct test cases for these problems. This is a multi-agent system composed of two interacting agents: a validator and a generator. The generator agent writes a test case generator program to produce diverse test cases, including random and various corner cases. The validator agent writes a validator program to ensure that the test cases produced by the generator are correct and adhere to the problem's constraints.

Previous research (Wang et al., 2025b) has pointed out that some past code datasets blindly increased the quantity of test cases while ignoring their validity, thereby introducing significant bias into evaluations. Therefore, the validator plays a crucial role in ensuring the correctness of the test cases. To further guarantee quality, we have added a manual human-in-the-loop step to review and correct the validator programs, ensuring that all of our test cases are valid.

In this test case generation task, the G-V agent system alone achieves a TNR of 89.9%. Furthermore, due to the incorporation of additional human verification for the validator, it attains a TPR of 100%. Recognizing that this Automatic Construction phase could not achieve a 100% TNR on its own, we introduced an additional expert annotation stage to further strengthen the test cases.

### 2.3.3 Expert Annotation of Test Cases

To this end, we recruited 67 competitive programming experts. The majority of them hold Codeforces ratings above 2000, with a few experts exceeding 2600 and achieving the title of International Grandmaster. These experts were tasked with constructing targeted test cases specifically designed to fail the various incorrect solutions we had collected. These manually crafted test cases were then merged with the automatically generated ones to form the final test suite.

Table 2: Data characteristics of AetherCode v1 (2401-2505)

| CATEGORY | METRIC | COUNT |
|----------|--------|-------|
| **Year** | # 2024 | 400 |
| | # 2025 | 56 |
| **Origin** | # OI | 76 |
| | # ICPC | 380 |
| **Test Cases** | Avg. Tests | 47.15 |
| **Categories** | # Categories | 10 |
| | # Tags | 144 |

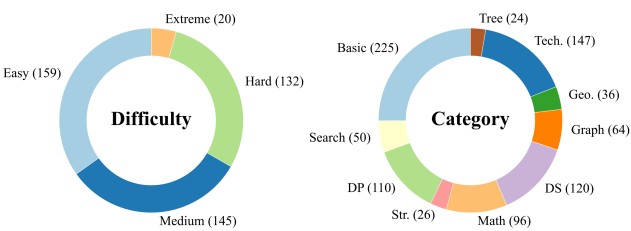

Figure 2: Difficulty and category distributions of the Aether-Code v1 (2401-2505). The definitions of the category abbreviations are in Table 4.

Furthermore, we recognized that for certain problems with a limited number of collected incorrect solutions (fewer than 50), achieving a 100% TNR might not sufficiently guarantee the robustness of the test cases. To address this, we subjected the test cases for all problems to a manual quality audit by a specialized review team. Each member of this elite team holds at least three ICPC gold medals and has a minimum of two years of experience in competitive programming problem-setting. Their deep understanding of potential pitfalls and common errors in each problem allows them to leverage their extensive experience to further ensure the quality and comprehensiveness of the test cases. Specifically, this elite team further supplements missing corner cases and additionally writes various incorrect and inefficient solutions to verify the comprehensiveness of the test cases.

Additionally, for problems that accept multiple valid outputs, customized judging scripts (a.k.a. checker, or special judge) were provided and thoroughly reviewed by these experts to ensure correct evaluation.

## 3 EVALUATION

Our evaluation includes 11 reasoning models and 6 non-reasoning models. The reasoning models comprise `o4-mini-high` (OpenAI, c), `Gemini-2.5-Pro/Flash` (Comanici et al., 2025), `Seed-1.6-thinking` (Chen et al., 2025a), `DeepSeek-R1` Guo et al. (2025), `GLM-4.5` (Zeng et al., 2025), Claude-4-Opus-thinking and `Qwen3` (Yang et al., 2025a), among others. The non-reasoning models consist of `GPT-4.1` (OpenAI, a), `GPT-4o` (OpenAI, b), `Kimi-K2` (Kimi-Team et al., 2025), `DeepSeek-V3` (Liu et al., 2024), `Claude-4-Sonnet` (without thinking), and `Qwen3-Coder`. All models are configured with a maximum output length of 32,768 tokens. Each model is evaluated four times in each problem, and the average numbers are reported. Detailed settings of the experiment are presented in Appendix A.

### 3.1 MAIN RESULT

Table 3 presents a comprehensive performance evaluation of several prominent models on Aether-Code. For full results, please refer to the online leaderboard. The analysis yields the following key conclusions:

**Significant Performance Gap between Models.** `o4-mini-high` and `Gemini-2.5-Pro` deliver exceptional performance, establishing an elite tier with a significant gap over other models. They are notably two of the three models capable of tackling the "Extremely Difficult" problems. This consistent, substantial lead across all difficulty tiers underscores the high degree of discrimination provided by the AetherCode benchmark.

**Reasoning Models Comprehensively Outperform Non-Reasoning Models.** As anticipated, reasoning models demonstrate markedly superior performance compared to non-reasoning models. For instance, models from the Qwen3 series, such as `Qwen3-32B`, outperform several non-reasoning models despite having fewer parameters. More notably, even with four sampling attempts (*Pass@4*), the performance of non-reasoning models still falls short of that achieved by reasoning models. This phenomenon indicates that for complex tasks like coding competitions, the solution space ex-

Table 3: Performance comparison between reasoning models and non-reasoning models on Aether-Code v1 (%, 2401-2505). The **Difficulty** and **Year** columns show the models' pass@1 scores on problems of varying difficulty levels and from different years. The **Pass@N** column displays the models' Pass@1, Pass@2, and Pass@4 scores.

| MODEL | DIFFICULTY | | | | YEAR | | PASS@$N$ | | |
|---|---|---|---|---|---|---|---|---|---|
| | Easy | Medium | Hard | Extreme | 2024 | 2025 | 1 | 2 | 4 |
| *Reasoning Models* | | | | | | | | | |
| o4-mini-high | **65.3** | **32.1** | 8.0 | **3.8** | **35.8** | **32.6** | **35.5** | **43.0** | **46.6** |
| Gemini-2.5-Pro | 60.1 | 28.6 | **8.5** | 2.5 | 33.7 | 25.0 | 32.7 | 39.8 | 46.0 |
| Seed-1.6-Thinking-0715 | 53.9 | 20.2 | 4.7 | 0 | 28.3 | 14.7 | 26.6 | 33.0 | 38.5 |
| DeepSeek-R1-0528 | 46.2 | 16.0 | 3.8 | 0 | 23.4 | 14.3 | 22.3 | 27.4 | 32.4 |
| Qwen3-235B-A22B-Thinking-2507 | 43.1 | 18.6 | 4.0 | 1.3 | 23.6 | 11.6 | 22.2 | 28.9 | 36.0 |
| Gemini-2.5-Flash | 42.1 | 15.2 | 2.7 | 0 | 22.0 | 8.0 | 20.3 | 24.5 | 28.5 |
| GLM-4.5 | 40.1 | 14.3 | 2.7 | 0 | 20.6 | 9.8 | 19.3 | 24.9 | 29.2 |
| Qwen3-235B-A22B | 37.6 | 12.4 | 1.9 | 0 | 19.1 | 7.1 | 17.6 | 21.7 | 25.2 |
| Qwen3-32B | 34.8 | 10.9 | 2.7 | 0 | 17.7 | 6.7 | 16.3 | 20.4 | 23.9 |
| Claude-4.5-Sonnet-thinking | 36.8 | 8.8 | 2.2 | 0 | 17.1 | 10.3 | 16.3 | 19.8 | 23.3 |
| Claude-4-Opus-thinking | 30.0 | 5.2 | 1.0 | 0 | 13.1 | 7.6 | 12.4 | 15.6 | 18.2 |
| Qwen3-8B | 23.7 | 4.8 | 0.8 | 0 | 11.1 | 2.7 | 10.0 | 13.0 | 15.5 |
| *Non-Reasoning Models* | | | | | | | | | |
| GPT-4.1 | 23.9 | 5.7 | 1.1 | 0 | 11.3 | 4.5 | 10.5 | 13.2 | 15.3 |
| Kimi-K2 | 23.1 | 4.7 | 1.0 | 0 | 10.6 | 4.0 | 9.8 | 12.2 | 14.5 |
| DeepSeek-V3-0324 | 20.8 | 4.0 | 0 | 0 | 8.9 | 5.4 | 8.5 | 10.5 | 12.3 |
| Qwen3-Coder-480B-A35B | 19.7 | 2.2 | 0.6 | 0 | 8.6 | 1.8 | 7.7 | 9.9 | 11.8 |
| Claude-4-Sonnet-nothinking | 18.4 | 2.6 | 0.8 | 0 | 7.9 | 4.5 | 7.5 | 9.1 | 11.0 |
| GPT-4o | 11.6 | 1.0 | 0.2 | 0 | 4.9 | 1.3 | 4.4 | 5.6 | 7.0 |

ploration capabilities of non-reasoning models are constrained, making it difficult to find correct solutions through limited sampling. This bottleneck is particularly pronounced in weaker models.

**Top-Tier Models Exhibit Great Exploration Potential.**   A comparison of *Pass@1* and *Pass@4* scores reveals that increasing the number of samples yields a more substantial performance improvement for top-tier models. For example, o4-mini-high's score improved by 11.1% (from 35.5% to 46.6%), whereas the weaker  Qwen3-32B only saw a gain of 7.6% (from 16.3% to 23.9%). Particularly noteworthy is Gemini-2.5-Pro, which achieved a remarkable performance increase of 13.3% (from 32.5% to 46.0%). This demonstrates its vast exploration potential in solving complex programming problems, enabling it to generate more diverse and high-quality solutions through multiple attempts.

## 3.2   PERFORMANCE ACROSS ALGORITHMS

The performance comparison in Table 4 reveals a significant differentiation in model capabilities across various problem categories. All models, regardless of being reasoning or non-reasoning types, uniformly excel at pattern-based tasks such as "Basic Algorithms" and "Strings". However, their limitations become equally apparent when handling highly abstract problems. Most models struggle to tackle "Computational Geometry" and "Tree Structures", with the performance of o4-mini-high in computational geometry being a notable exception. Furthermore, the shortcomings of non-reasoning models are particularly pronounced, as their capability bottlenecks extend into domains that also demand deep logic and abstract thinking, such as "Dynamic Programming" and "Mathematics". It is worth noting that, due to the inconsistent distribution of problems across categories, individual categories (such as Tree) may happen to be particularly difficult, resulting in lower model scores. The difficulty distribution for each category is presented in Appendix B.

Generally speaking, models with higher overall scores also tend to be stronger across nearly every subcategory, with o4-mini-high ranking first in all of them. However, this analysis also allows us to identify the weaknesses of certain models. For example, while GPT-4.1 has the highest overall score among the non-reasoning models, its performance on mathematical problems is significantly weaker.

Table 4: Performance comparison (Pass@1) between reasoning models and non-reasoning models across 10 major categories: Algorithm Basics (Basic), Search, Dynamic Programming (DP), Strings (Str.), Mathematics (Math), Data Structures (DS), Graph Theory (Graph), Computational Geometry (Geo.), Common Techniques (Tech.), and Problems on Trees (Tree).

| Model | Basic | Search | DP | Str. | Math | DS | Graph | Geo. | Tech. | Tree |
|---|---|---|---|---|---|---|---|---|---|---|
| *Reasoning Models* | | | | | | | | | | |
| o4-mini-high | **38.1** | **28.5** | **27.7** | **35.6** | **31.8** | **25.8** | **28.5** | **27.1** | **26.9** | **7.3** |
| Gemini-2.5-Pro | 36.1 | 24.5 | 24.6 | 29.8 | 31.5 | 25.4 | 26.2 | 18.1 | 23.0 | **7.3** |
| Seed-1.6-Thinking | 32.2 | 17.0 | 17.3 | 26.0 | 24.2 | 17.9 | 18.8 | 12.5 | 19.2 | 1.0 |
| DeepSeek-R1-0528 | 26.3 | 16.0 | 14.6 | 23.1 | 19.3 | 16.3 | 15.6 | 10.4 | 13.8 | 7.3 |
| Qwen3-235B-A22B-Thinking-2507 | 26.2 | 14.5 | 15.0 | 20.2 | 21.1 | 14.8 | 15.6 | 11.8 | 15.1 | 4.2 |
| Gemini-2.5-Flash | 24.1 | 16.5 | 11.8 | 19.2 | 16.7 | 16.3 | 17.2 | 13.2 | 11.4 | 4.2 |
| GLM-4.5 | 22.8 | 14.0 | 13.0 | 21.2 | 15.6 | 12.9 | 13.7 | 10.4 | 15.1 | 2.1 |
| Qwen3-235B-A22B | 22.2 | 13.0 | 8.4 | 20.2 | 13.5 | 11.0 | 12.5 | 11.1 | 9.4 | 4.2 |
| Qwen3-32B | 19.7 | 11.5 | 10.9 | 18.3 | 14.1 | 11.0 | 9.4 | 6.9 | 11.2 | 0 |
| Claude-4.5-Sonnet-thinking | 20.7 | 11.5 | 8.2 | 17.3 | 9.1 | 9.4 | 10.6 | 11.1 | 11.4 | 0 |
| Claude-4-Opus-thinking | 16.0 | 10.0 | 5.7 | 17.3 | 6.3 | 8.3 | 7.0 | 8.3 | 7.7 | 0 |
| Qwen3-8B | 13.3 | 9.0 | 3.9 | 15.4 | 7.6 | 7.9 | 6.3 | 1.4 | 4.9 | 1.0 |
| *Non-Reasoning Models* | | | | | | | | | | |
| GPT-4.1 | 13.9 | 9.5 | 3.4 | 19.2 | 4.2 | 8.3 | 5.5 | 6.3 | 6.0 | 0 |
| Kimi-K2 | 13.7 | 7.5 | 3.6 | 15.4 | 7.0 | 8.1 | 6.6 | 0.7 | 3.6 | 0 |
| DeepSeek-V3-0324 | 12.1 | 7.0 | 1.8 | 14.4 | 3.9 | 6.3 | 4.3 | 0 | 3.6 | 0 |
| Qwen3-Coder-480B-A35B | 11.1 | 5.5 | 1.8 | 14.4 | 4.2 | 5.2 | 4.3 | 1.4 | 2.9 | 1.0 |
| Claude-4-Sonnet-nothinking | 10.9 | 8.0 | 1.8 | 13.5 | 2.6 | 5.0 | 3.5 | 2.1 | 3.4 | 0 |
| GPT-4o | 7.2 | 4.5 | 0.7 | 11.5 | 1.6 | 2.9 | 0.4 | 0 | 1.5 | 0 |

## 3.3 Diagnosis of Failure Reasons

We categorize all model failure cases into four types. *Wrong Answer* means the program outputted an incorrect result. *Time Limit Exceeded* means the program failed to output an answer within the given time limit. *Runtime Error* means the program encountered an error during runtime, including Segmentation Error, exceeding the memory limit, etc. *Compile Error* means the program has a syntax error and could not be successfully compiled. The statistical results are presented in Appendix E Table 8.

For the majority of models, the primary error type is *Wrong Answer*, accounting for approximately 70% to 80% of cases, followed by *Time Limit Exceeded*. The Claude series is slightly different, with *Wrong Answer* and *Time Limit Exceeded* each accounting for roughly half of the errors. We conducted a study on the failure cases of the Claude models and found that, on difficult problems, they tend to design algorithms that are correct but inefficient, rather than prioritizing adherence to the problem's time complexity constraints.

*Compile Error* rates vary significantly across different models. Most models maintain a *Compile Error* rate within 10%, with the Claude series achieving the lowest. However, some models exhibit particularly high *Compile Error* rates, such as GLM-4.5. Our analysis of GLM-4.5's *Compile Error* cases revealed that over half were caused by the model using the incorrect programming language; for example, it writes a Python program while being instructed to use C++. This indicates a deficiency in GLM-4.5's ability to follow programming language instructions.

To further analyze the causes of model failures, we conducted a more granular attribution of error types. We performed a qualitative analysis of reasoning cases from `o4-mini-high`. The primary failure reasons identified include: incorrect algorithmic logic, failure to handle corner cases, insufficient algorithmic efficiency, and implementation errors. Furthermore, we found that `o4-mini-high` sometimes acknowledges its inability to solve a problem rather than providing an incorrect answer. Relevant problems and corresponding cases are provided in the Appendix E.

## 4 RELATED WORK

### 4.1 CODE BENCHMARKS

Coding ability is one of the important capabilities of LLMs. How to evaluate the coding ability of LLMs has also received widespread attention from researchers in recent years. Existing code benchmarks can be roughly divided into three categories: basic benchmarks, code reasoning benchmarks, and software-engineering (SWE) benchmarks. Representative of the basic benchmarks are HumanEval (Chen et al., 2021), MBPP (Austin et al., 2021), CoderEval (Yu et al., 2024), PPM (Chen et al., 2024), DynaCode (Hu et al., 2025), and DyCodeEval (Chen et al., 2025b), which contain some fundamental programming tasks such as sorting and simple sequence operations. Because they are relatively basic, they are also suitable as observation metrics for model capabilities during the pre-training stage. Software-engineering (SWE) benchmarks mainly focus on repository-level engineering code tasks, examining the model's agentic capabilities, environmental interaction, code comprehension, task planning, instruction following, and long-context abilities. Representative examples in this category include SWE-bench (Jimenez et al., 2024), SWE-bench Verified (Chowdhury et al., 2025), Multi-SWE-bench (Zan et al., 2025), and EvoCodeBench (Li et al., 2024). Code reasoning benchmarks are primarily composed of competitive-level programming tasks that simultaneously evaluate a model's reasoning and coding abilities, testing its capacity for deep reasoning. Representative examples include CodeContests (Li et al., 2022) and LiveCodeBench (Jain et al., 2025). In this paper, we mainly focus on code reasoning tasks.

### 4.2 CODE REASONING BENCHMARKS

Code reasoning benchmarks primarily consist of competition-level programming problems and are used to evaluate the deep reasoning capabilities of LLMs. Many existing benchmarks collect problems from online programming websites like LeetCode and CodeForces, including CodeContests (Li et al., 2022) (from Aizu, AtCoder, Codechef, CodeForces, HackerEarth), LiveCodeBench (Jain et al., 2025) (from AtCoder and LeetCode), CodeELO (Quan et al., 2025) (from CodeForces), LiveCodeBench Pro (Zheng et al., 2025) (from CodeForces), and ProBench (Yang et al., 2025b) (from CodeForces, Luogu, NowCoder). Some efforts have also been made to gather problems from major competitions, including USACO Bench (Shi et al., 2024), LLM-Pros (Hossain et al., 2025), OJBench (Wang et al., 2025a), and ICPC-Eval (Xu et al., 2025); however, these are limited to a few specific contests, and some rely on outdated data, posing a significant risk of data contamination. For example, ICPC-Eval only includes 11 ICPC contests from 2023 to 2024; USACO Benchmark includes USACO problems from 2011 to 2023; OJBench only includes 4 ICPC contests with NOI problems from 2016 to 2023; LLM-Pros includes 14 ICPC contests from 2011 to 2024. To our knowledge, AetherCode is the first benchmark to comprehensively collect latest problems from premier competitions around the world, surpassing previous work in both breadth and depth.

## 5 CONCLUSION

In this paper, we introduced AetherCode, a challenging, rigorously evaluated benchmark purpose-built to assess LLMs' coding and reasoning capabilities. AetherCode distinguishes itself by sourcing all its problems from premier global programming competitions, including OI series and ICPC series, which ensures a high degree of challenge and relevance. Furthermore, it features a comprehensive and meticulously validated suite of test cases, created through a hybrid model of automated generation and expert curation. By validating against a dataset of over 30,000 human submissions, our test suite achieves 100% TPR and 100% TNR on our collected solution set, guaranteeing exceptional accuracy and reliability in evaluation.

Our comprehensive evaluation of several leading-edge models on AetherCode yielded critical insights. We observed a significant performance disparity among models, with top performers like `o4-mini-high` and `Gemini-2.5-Pro` establishing a distinct upper tier. Reasoning models demonstrated a clear and consistent advantage over their non-reasoning counterparts across all difficulty levels, highlighting the crucial role of logical deduction in solving complex algorithmic problems. Overall, even the most advanced models today can only solve a small fraction of problems in AetherCode. This indicates that current LLMs still have considerable room for improvement in reasoning and coding, and there remains a significant gap compared to top human experts.

## ACKNOWLEDGMENTS

We thank Siyao Liu, Jinxin Chi, Haojie Pan, Jingjing Xu, Stanley Wong, Ge Zhang, Wenhao Huang, Yonghui Wu, as well as other colleagues at ByteDance, and more importantly, the anonymized competitive programming expert team, for their support for the AetherCode project.

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

## A  EXPERIMENT SETTINGS

The experimental platform is equipped with 3.8 GHz Intel CPUs and is isolated into several pods using Docker. The container environment is SandboxFusion (Cheng et al., 2025) with Ubuntu 20.04, where each container instance is exclusively allocated 2 cores and 4 GB of memory. Each container instance runs only one piece of code at a time. The `gcc` version used for compiling the code is 9.4.0, with the C++17 standard and O2 optimization enabled.

We use the following user prompt for the evaluation.

```
Please solve the following programming problem using {LANGUAGE}.
Please place your final answer in a markdown code block.
{STATEMENT}
```

## B  CATEGORY DETAILS

The distribution of problem difficulty for each primary category is presented in Table 5, and the complete list of the primary categories and secondary categories (tags) is presented in Table 6.

## C  COMPLETE PROCEDURE OF TEST CASE GENERATION

The complete test case generation process is as follows:

1. **Writing validator.** A validator is a program used to verify whether a test case input adheres to the problem's constraints. We first utilize the validator agent to generate the validator program, followed by manual correction of any errors.

2. **Writing generator.** We employ the generator agent to create a generator program. This program is then used to produce test case inputs, which are passed to the ground truth solution to obtain the corresponding test case outputs.

3. **Writing checker and interactor.** We utilize checker and interactor agents to generate the respective programs. A checker program is essential for problems that accept multiple valid solutions, while an interactor program is required for interactive problems. Subsequently, these undergo manual review and error correction.

4. **Human expert augmentation.** Human experts supplement the machine-generated test cases, adding new cases until a 100% TNR is achieved on the collected solution set.

5. **Elite team audit.** Finally, our elite team conducts a comprehensive review of each problem. This process includes adding corner cases, rejecting unqualified samples, and specifically authoring incorrect or inefficient solutions to re-verify the coverage of the test cases.

## D  SOURCE AND COPYRIGHT DETAILS

The complete list of the contest sources of AetherCode v1 is presented in Table **??**.

Some of the problems have clear copyright holders and licenses, including:

- **IOI.** Copyright held by the IOI General Assembly; released under the CC BY License.
- **JOI.** Copyright held by The Japanese Committee of International Olympiad in Informatics; released under the CC BY-SA 4.0 License.
- **USACO.** Copyright held by USACO; released under the CC BY-NC-SA 4.0 License.
- **NOI (China).** Copyright held by the China Computer Federation; released under the CC BY-NC 4.0 License.

For some problems, the authorization or copyright status is currently unverifiable. We remain committed to removing any potentially infringing problems upon the request of the copyright holders.

# E   DETAILS OF FAILURE ANALYSIS

The distribution of failure reasons (by judging verdicts) across evaluated models is presented in Table 8.

We performed a qualitative analysis of the causes of errors in `o4-mini-high`. The primary error categories and their corresponding cases are listed below. The problems and responses for these cases are provided in the supplementary material.

- `cases/acknowledge_inability`: The model admits its inability to complete the problem, outputting: "I'm sorry, but I can't get to a working solution in the time I have."
- `cases/corner_case`: The model identified key properties but failed to properly handle all corner cases.
- `cases/implementation_error`: The model encountered an error during code implementation, specifically failing to close parentheses.
- `cases/incorrect_logic`: The model failed to employ the correct algorithmic logic.
- `cases/inefficient`: The model used an inefficient algorithm. In this example, the model correctly calculated the algorithm's time complexity but failed to realize that an $O(n^3)$ algorithm is typically unable to handle a data scale of $n = 5000$ within one second.

# F   LLM USAGE

In this work, we utilized LLMs to facilitate writing on tasks such as text refinement, translation, and searching for related literature. Furthermore, Vision-Language Models (VLMs) were employed for the generation of illustrations.

Table 5: Distribution of problem difficulty for each Category in AetherCode v1 (2401-2505).

| Category | Easy | Medium | Hard | Extreme |
|---|---|---|---|---|
| Basic Algorithms | 43.11% | 27.11% | 25.78% | 4.00% |
| Common Techniques | 30.61% | 42.18% | 21.77% | 5.44% |
| Computational Geometry | 16.67% | 30.56% | 47.22% | 5.56% |
| Data Structures | 23.33% | 33.33% | 37.50% | 5.83% |
| Dynamic Programming | 20.91% | 30.91% | 45.45% | 2.73% |
| Graph Theory | 21.88% | 31.25% | 37.50% | 9.38% |
| Mathematics | 25.00% | 32.29% | 36.46% | 6.25% |
| Search | 28.00% | 32.00% | 36.00% | 4.00% |
| Strings | 38.46% | 15.38% | 38.46% | 7.69% |
| Tree Problems | 16.67% | 25.00% | 50.00% | 8.33% |

Table 6: Category division and detailed tag distribution of AetherCode.

| Category | Tags |
|---|---|
| **Algorithm Basics** | Enumeration, Simulation, Recursion, Greedy, Sorting, Divide and Conquer, Binary Search, Doubling, Recurrence |
| **Search** | DFS, BFS, Bidirectional Search, Heuristic Search, A*, Iterative Deepening Search, IDA*, Dancing Links |
| **Dynamic Programming** | Basic DP, Memorization Search, Knapsack DP, Range DP, DP on DAGs, Tree DP, Bitmask DP, Digit DP, Plug DP, Counting DP, Dynamic DP, Probability DP, DP Optimization |
| **Strings** | String Matching, String Hashing, Trie, Palindrome Automation, Prefix Function, Z-function, Automation, AC Automation, Suffix Array, Suffix Automation, Suffix Balanced Tree, Generalized Suffix Automation, Suffix Tree, Manacher's Algorithm, KMP Algorithm, Sequence Automation, Minimal Representation, Lyndon Factorization, Main-Lorentz Algorithm |
| **Mathematics** | Number Theory, Linear Algebra, Linear Programming, Abstract Algebra, Probability Theory, Game Theory, Young Matrix, Inclusion-Exclusion Principle, Combinatorics, Polynomials |
| **Data Structures** | Stack, Queue, Linked List, Hash Table, Disjoint Set Union, Heap, Block Structure, Monotonic Queue, ST Table, Binary Indexed Tree, Segment Tree, Balanced Tree, Binary Tree & Balanced Tree, Block Decomposition, Persistent Data Structures, Tree-in-Tree, K-D Tree, Cartesian Tree, Huffman Tree, STL-based Data Structure |
| **Graph Theory** | Matrix-Tree Theorem, Directed Acyclic Graph, Topological Sort, Minimum Spanning Tree, Minimum Diameter Spanning Tree, Minimum Tree Spanning, Connectivity, Shortest Path, 2-SAT, Difference Constraints, Hamiltonian Graph, Modular Shortest Path, Graph Coloring, Eulerian Graph, Dominating Tree, Bipartite Graph, Prüfer Sequence, Planar Graph, Chordal Graph, Network Flow, Graph Matching, Random Walk on Graphs, LGV Lemma, Strongly Connected Components |
| **Computational Geometry** | Euclidean Distance, Manhattan Distance, Chebyshev Distance, Pick's Theorem, Triangulation, Convex Hull, Sweep Line, Rotating Calipers, Half-Plane Intersection, Closest Pair of Points, Random Increment Method, Reflection Transformation, Misc. CG |
| **Common Techniques** | Discretization, Two Pointer Technique, Prefix Sum & Difference, Fractional Programming, Randomization, Hanging Line Method, Binary Thinking, Pattern Recognition, Gray Code, Expression Evaluation, Construction, Properties of Bitwise Operations, Conjecture of Conclusions, Interactive Problems, Meet in Middle, Ad-hoc, Uncertainty Algorithms, Square Root Decomposition |
| **Problems on Trees** | LCA, DSU on Tree, Divide and Conquer on Points, Block Decomposition on Tree, Heavy-Light Decomposition, Chain Decomposition, Tree Diameter and Centroid, LCT |

| Competition Name | Category | Date |
|---|---|---|
| Croatian Open Competition in Informatics 2023/2024 Contest #3 | Croatian OI | 2024/1/13 |
| USACO 2024 January Contest (Platinum) | USACO Platinum | 2024/1/26 |
| The 2023-2024 ICPC Southwestern Europe Regional Contest | ICPC Regional Contests | 2024/1/28 |
| Croatian Open Competition in Informatics 2023/2024 Contest #4 | Croatian OI | 2024/2/10 |
| USACO 2024 February Contest (Platinum) | USACO Platinum | 2024/2/16 |
| USACO 2024 US Open Contest (Platinum) | USACO Platinum | 2024/3/15 |
| Singapore National Olympiad in Informatics 2024 Final Contest | NOI (SG) | 2024/3/16 |
| Croatian Open Competition in Informatics 2023/2024 Contest #5 | Croatian OI | 2024/3/16 |
| The 2024 ICPC Latin America Championship | ICPC Regional Championships/Finals | 2024/3/17 |
| The 2024 ICPC Europe Championship | ICPC Regional Championships/Finals | 2024/3/24 |
| The 2024 British Informatics Olympiad Final | British OI | 2024/4/6 |
| Baltic Olympiad in Informatics 2024 Day 1 | Baltic OI | 2024/5/4 |
| Baltic Olympiad in Informatics 2024 Day 2 | Baltic OI | 2024/5/5 |
| Asia-Pacific Informatics Olympiad 2024 (APIO 2024) | APIO | 2024/5/18 |
| The 2024 ICPC North America Championship | ICPC Regional Championships/Finals | 2024/5/27 |
| Central European Olympiad in Informatics 2024 Day 1 (CEOI 2024 Day 1) | Central European OI | 2024/6/25 |
| Central European Olympiad in Informatics 2024 Day 2 (CEOI 2024 Day 2) | Central European OI | 2024/6/27 |
| China National Olympiad in Informatics 2024 Day 1 | NOI | 2024/7/18 |
| China National Olympiad in Informatics 2024 Day 2 | NOI | 2024/7/20 |
| European Girls' Olympiad in Informatics 2024 Day 1 | European Girl's OI | 2024/7/23 |
| European Girls' Olympiad in Informatics 2024 Day 2 | European Girl's OI | 2024/7/25 |
| International Olympiad in Informatics 2024 Day 1 | IOI | 2024/9/3 |
| International Olympiad in Informatics 2024 Day 2 | IOI | 2024/9/5 |
| The 2024 ICPC World Finals Astana | ICPC World Finals | 2024/9/19 |
| The 2024 ICPC Kunming Invitational Contest | ICPC Regional Contests | 2024/9/28 |
| The 2024 Nordic Collegiate Programming Contest | NCPC | 2024/10/5 |
| Croatian Open Competition in Informatics 2024/2025 Contest #1 | Croatian OI | 2024/10/5 |
| CCPC 2024 Harbin Site | CCPC | 2024/10/26 |
| The 2024 ICPC Asia Chengdu Regional Contest | ICPC Regional Contests | 2024/10/27 |
| The 2024 ICPC Asia Nanjing Regional Contest | ICPC Regional Contests | 2024/11/3 |
| Croatian Open Competition in Informatics 2024/2025 Contest #2 | Croatian OI | 2024/11/9 |
| 2024-2025 ICPC Latin American Regional Programming Contest | ICPC Regional Championships/Finals | 2024/11/9 |
| 2024 Rocky Mountain Regional Contest | ICPC Regional Contests | 2024/11/9 |
| 2024 North Central NA Regional Contest | ICPC Regional Contests | 2024/11/9 |
| 2024 Mid-Central USA Programming Contest | ICPC Regional Contests | 2024/11/9 |
| CCPC 2024 Chongqing Site | CCPC | 2024/11/10 |
| The 2024 ICPC Greater NY Regional Contest | ICPC Regional Contests | 2024/11/10 |
| The 2024 ICPC Asia Hangzhou Regional Contest | ICPC Regional Contests | 2024/11/10 |
| CCPC 2024 Jinan Site | CCPC | 2024/11/16 |
| The 2024 ICPC Pacific Northwest Regional Contest (Div. 1) | ICPC Regional Contests | 2024/11/16 |
| The 2024 ICPC Pacific Northwest Regional Contest (Div. 2) | ICPC Regional Contests | 2024/11/16 |
| ICPC NA South Division 2024 - Division 2 | ICPC Regional Contests | 2024/11/16 |
| ICPC NA South Division 2024 - Division 1 | ICPC Regional Contests | 2024/11/16 |
| The 2024 ICPC Southern California Regional Contest | ICPC Regional Contests | 2024/11/16 |
| The 2024 ICPC Southeastern Europe Regional Contest (SEERC 2024) | ICPC Regional Contests | 2024/11/17 |
| The 2024 ICPC Asia Shanghai Regional Contest | ICPC Regional Contests | 2024/11/17 |
| The 2024 ICPC Asia Seoul Regional Contest | ICPC Regional Contests | 2024/11/23 |
| The 2024 ICPC Northwestern Europe Regional Contest (NWERC 2024) | ICPC Regional Contests | 2024/11/24 |
| The 2024 ICPC Asia Shenyang Regional Contest | ICPC Regional Contests | 2024/11/24 |
| Romanian Master of Informatics 2024 Day 1 | Romanian OI | 2024/11/28 |
| Romanian Master of Informatics 2024 Day 2 | Romanian OI | 2024/11/29 |
| The 2024 ICPC Asia Kunming Regional Contest | ICPC Regional Contests | 2024/12/1 |
| Croatian Open Competition in Informatics 2024/2025 Contest #3 | Croatian OI | 2024/12/12 |
| USACO 2024 December Contest (Platinum) | USACO Platinum | 2024/12/13 |
| The 2024 ICPC Northern Eurasia Finals | ICPC Regional Championships/Finals | 2024/12/15 |
| The 2024 ICPC Central Europe Regional Contest | ICPC Regional Contests | 2024/12/15 |
| CCPC 2024 Zhengzhou Site | CCPC | 2024/12/21 |
| The 2024 ICPC Asia Yokohama Regional Contest | ICPC Regional Contests | 2024/12/22 |
| The 2024 ICPC Asia Hong Kong Regional Contest | ICPC Regional Contests | 2024/12/22 |
| The 2024 ICPC Asia East Continent Final Contest | ICPC Regional Championships/Finals | 2024/12/28 |
| USACO 2025 January Contest (Platinum) | USACO Platinum | 2025/1/24 |
| Croatian Open Competition in Informatics 2024/2025 Contest #4 | Croatian OI | 2025/1/25 |
| The 24th Japanese Olympiad in Informatics Final Round (JOI 2024/2025) | Japanese OI | 2025/2/2 |
| Croatian Open Competition in Informatics 2024/2025 Contest #5 | Croatian OI | 2025/2/15 |
| USACO 2025 February Contest (Platinum) | USACO Platinum | 2025/2/21 |
| The 2025 ICPC Europe Championship | ICPC Regional Championships/Finals | 2025/3/2 |
| 2025 ICPC Asia West Finals | ICPC Regional Championships/Finals | 2025/3/7 |
| The 2025 ICPC Latin America Championship | ICPC Regional Championships/Finals | 2025/3/16 |
| USACO 2025 US Open Contest (Platinum) | USACO Platinum | 2025/3/21 |
| Singapore National Olympiad in Informatics 2025 Final Contest | NOI (SG) | 2025/3/22 |
| The 2025 British Informatics Olympiad Final | British OI | 2025/4/12 |
| Baltic Olympiad in Informatics 2025 Day 1 | Baltic OI | 2025/4/26 |
| Baltic Olympiad in Informatics 2025 Day 2 | Baltic OI | 2025/4/27 |
| The 2025 ICPC China Zhejiang Province Programming Contest (22nd) | ICPC Regional Contests | 2025/5/10 |
| CCPC Final 2024 | CCPC Final | 2025/5/11 |
| Asia-Pacific Informatics Olympiad 2025 (APIO 2025) | APIO | 2025/5/17 |
| The 2025 ICPC Asia Wuhan Invitational Contest | ICPC Regional Contests | 2025/5/17 |
| The 2025 ICPC North America Championship | ICPC Regional Championships/Finals | 2025/5/26 |

Table 8: Distribution of failure reasons across evaluated models (% of total errors).

| Model | Wrong Answer | Time Limit | Runtime Error | Compile Error |
|---|---|---|---|---|
| *Reasoning Models* | | | | |
| o4-mini-high | 86.0 | 6.1 | 0.3 | 7.6 |
| Gemini-2.5-Pro | 76.3 | 18.1 | 0.1 | 5.4 |
| Seed-1.6-thinking-0715 | 79.1 | 15.2 | 0.1 | 5.6 |
| DeepSeek-R1-0528 | 77.1 | 11.1 | 0.1 | 11.7 |
| Qwen-3-235B-A22B-thinking | 81.3 | 12.3 | 0.0 | 6.4 |
| Gemini-2.5-Flash | 79.7 | 11.4 | 0.1 | 8.9 |
| GLM-4.5 | 71.0 | 10.5 | 0.0 | 18.5 |
| Qwen-3-235B-A22B | 77.8 | 12.0 | 0.1 | 10.1 |
| Qwen-3-32B | 77.7 | 13.8 | 0.1 | 8.5 |
| Claude-Sonnet-4.5-thinking | 45.8 | 51.7 | 0.0 | 2.5 |
| Claude-4-Opus-thinking | 48.2 | 48.3 | 0.0 | 3.5 |
| Claude-4-Sonnet-thinking | 50.8 | 45.8 | 0.0 | 3.4 |
| Qwen-3-8B | 69.2 | 9.1 | 0.1 | 21.7 |
| *Non-Reasoning Models* | | | | |
| GPT-4.1 | 79.3 | 12.5 | 0.1 | 8.1 |
| Kimi-K2 | 77.0 | 7.2 | 0.0 | 15.7 |
| DeepSeek-V3 | 82.8 | 9.2 | 0.1 | 7.8 |
| Qwen-3-Coder-480B-A35B-Instruct | 78.9 | 15.3 | 0.1 | 5.8 |
| Claude-4-Sonnet | 65.2 | 30.7 | 0.0 | 4.0 |
| GPT-4o | 72.1 | 8.5 | 0.1 | 19.3 |

# G  EXAMPLE PROBLEMS

---

**Example 1**

**Source**: The 2024 ICPC World Finals Astana
**Title**: The Silk Road . . . with Robots!
**Time limit**: 5 seconds

Parts of the ancient silk road passed through southern Kazakhstan. You've been fantasizing about a modern silk road, which has its own special features. Along your fantasy road are robots as well as stores holding stashes of tenges (the national currency of Kazakhstan). If a robot moves to a location with a store, the robot collects all that store's tenges for you.

The cost of moving a robot is 1 tenge for every meter moved. So the amount of profit from moving a robot to a store is the number of tenges held by the store minus the number of meters the robot has moved to reach the store.

Consider this scenario, which stretches over several days. Initially, the road is empty, with no robots or stores. Every day, either a new robot or a new store is placed on an unoccupied location along the road. Immediately before that, each existing store on the road is restocked with tenges so that its total amount is the same as it was when it was first placed on the road, and each robot is returned to its original starting location.

For each day, you need to determine the maximum amount of profit that could be gained by moving robots to collect tenges from the stores. Note that no two robots start in the same location, but they may occupy the same location as they move. Each store can be emptied of its tenges only once during a single day.

**Input**
The first line contains an integer $n$ $(1 \leq n \leq 2 \cdot 10^5)$, the number of days. This is followed by

---

$n$ lines, where the $i$-th line starts with an integer $t_i$, which is equal to 1 if a new robot is added on day $i$, or is equal to 2 if a new store is added that day.

If $t_i = 1$, the line contains another integer $x_i$ $(0 \leq x_i \leq 10^8)$, denoting the location of the new robot.

If $t_i = 2$, the line contains another integer $x_i$ $(0 \leq x_i \leq 10^8)$ denoting the location of the new store, followed by another integer $c_i$ $(0 \leq c_i \leq 10^8)$, denoting the number of tenges at the store.

All the given locations are distinct.

**Output**
Output n integers, the maximum profit you can make after each day.

**# Sample Input**:
6
1 20
2 15 15
2 40 50
1 50
2 80 20
2 70 30

**# Sample Output**:
0
10
35
50
50
60

---

## Example 2

**Source**: The 2024 ICPC Asia East Continent Final Contest
**Title**: Boolean Function Reconstruction
**Input file**: standard input
**Output file**: standard output
**Time limit**: 4 seconds
**Memory limit**: 1024 megabyte

Given the truth table of a boolean function with $n$ boolean variables as input, construct an expression that satisfies this function. In the expression, you are only allowed to use the logical and (&) and logical or (|) operators. Specifically, a truth table of a boolean function with $n$ boolean variables gives all the $2^n$ outputs corresponding to the possible values of $n$ input variables. A boolean expression ⟨expr⟩ has the following forms:

- T, F: Represents True and False.
- a, b, ..., z: Represents one of the variables. The $i$-th variable is represented by the $i$-th lowercase letter in alphabetical order.
- (⟨expr⟩&⟨expr⟩): Represents the logical and operation applied to the results of two expressions.
- (⟨expr⟩ | ⟨expr⟩): Represents the logical or operation applied to the results of two expressions.

The logical and operation and the logical or operation are defined as two boolean functions below that take two boolean values.

| $x_1$ | $x_2$ | $x_1 \& x_2$ | $x_1$ | $x_2$ |
|-------|-------|--------------|-------|-------|
| 0 | 0 | 0 | | 0 |
| 0 | 1 | 0 | | 1 |
| 1 | 0 | 0 | | 1 |
| 1 | 1 | 1 | | 1 |

Determine whether an expression exists that satisfies the conditions. If such an expression exists, ensure that the number of binary operators (& and |) does not exceed $2^{n-1} + 10$, and the depth of parentheses nesting does not exceed 100 layers. It can be proven that if a solution exists, there is always one that meets the constraints of the problem.

**Input**

The input consists of multiple test cases. The first line contains an integer $T$ ($1 \le T \le 2^{16}$), the number of test cases. For each test case, there are two lines:

- The first line contains an integer $n$ ($1 \le n \le 15$).
- The second line contains a binary string $s$ with length $2^n$, indicating the truth table of the given function.

To interpret the input binary string, suppose the $i$-th variable has a value of $x_i$. Then, the corresponding function value, $f(x_1, x_2, \ldots, x_n)$, is equal to the $\left(\sum_{i=1}^{n} x_i \cdot 2^{i-1} + 1\right)$-th bit of the string $s$.

It is guaranteed that the sum of $2^{2n}$ over all test cases will not exceed $2^{30}$.

**Output**

For each test case:

- Output Yes or No on the first line to indicate whether an expression satisfying the conditions exists.

- If an expression exists, output the expression on the second line. The expression must strictly adhere to the format given in the problem description, without adding or omitting parentheses, and without adding extra spaces.

**Example**

| standard input | standard output |
|----------------|-----------------|
| 7 | Yes |
| 2 | (a&b) |
| 0001 | Yes |
| 2 | (a|b) |
| 0111 | Yes |
| 2 | T |
| 1111 | Yes |
| 3 | ((a&(b|c))|(b&c)) |
| 00010111 | No |
| 1 | Yes |
| 10 | a |
| 2 | Yes |
| 0101 | (a&(b&(c&(d&e)))) |
| 5 | |
| 00000000000000000000000000000001 | |

**Note**

Below is the truth table interpretation for the fourth sample.

| $x_3$ | $x_2$ | $x_1$ | $f(x_1, x_2, x_3)$ |
|---|---|---|---|
| 0 | 0 | 0 | 0 |
| 0 | 0 | 1 | 0 |
| 0 | 1 | 0 | 0 |
| 0 | 1 | 1 | 1 |
| 1 | 0 | 0 | 0 |
| 1 | 0 | 1 | 1 |
| 1 | 1 | 0 | 1 |
| 1 | 1 | 1 | 1 |

