# OpenReview forum: "AetherCode: Evaluating LLMs’ Ability to Win In Premier Programming Competitions"
_ICLR.cc/2026/Conference — ICLR 2026 Poster_

### Official Review · Reviewer_eg8w · 2025-10-19

**Soundness:** 3
**Presentation:** 3
**Contribution:** 2
**Rating:** 2
**Confidence:** 4

**Summary:**

This paper introduces AetherCode, a new benchmark for evaluating the code reasoning capabilities of Large Language Models (LLMs). The authors argue that existing benchmarks are insufficient due to limited difficulty and scope, as well as evaluation biases from low-quality test cases. To address this, AetherCode is constructed using problems from premier programming competitions like the International Olympiad in Informatics (OI) and the International Collegiate Programming Contest (ICPC).

Specifically, the authors collect 456 challenging problems from world-class competitions, processed into a unified Markdown+LaTeX format. A novel hybrid approach for test case generation is proposed, combining an automated Generator-Validator (G-V) agent with intensive human expert annotation. The quality is assessed using a True Positive Rate (TPR) and True Negative Rate (TNR) metric against a large set of human solutions, reportedly achieving 100% on both.

**Strengths:**

1. The paper's core strength lies in sourcing problems from truly top-tier competitions (IOI, ICPC World Finals, etc.). This is a commendable step up in difficulty and complexity compared to benchmarks sourced primarily from online judges like LeetCode or CodeForces.
2. The most significant contribution is the meticulous attention paid to test case quality. The proposed methodology of evaluating the entire test suite as a binary classifier using TPR and TNR on a large corpus of human submissions is a novel and valuable idea for the community. Achieving 100% TPR/TNR on their collected set is an impressive feat of engineering.
3. The detailed curation pipeline, from PDF conversion to multi-dimensional expert categorization (difficulty, algorithm tags), is thorough and sets a high standard for future benchmark development (Figure 1)

**Weaknesses:**

1. The paper claims AetherCode is the first benchmark to systematically collect problems from premier programming competitions worldwide. This claim is overstated. The paper itself cites several recent works with similar goals, such as USACO Bench, OJBench, and ICPC-Eval, but fails to adequately differentiate itself. Merging the data sources from several existing benchmarks and stating the comprehensiveness should not be a significant contribution to me. A more nuanced discussion is needed to pinpoint AetherCode's unique value proposition beyond just scale.

2. While the TPR/TNR metric is a great idea, the claim of achieving a 100% True Negative Rate (TNR) is only as strong as the set of incorrect solutions it was tested against.
- 2.1 The paper provides little detail on the nature of the 30,000+ incorrect human submissions. Do they include solutions with subtle, hard-to-find bugs (e.g., off-by-one errors, mishandled edge cases, time-limit-exceeded on specific inputs), or are they mostly solutions that fail on simple cases?
- 2.2 A 100% TNR on a potentially limited set of known failure modes does not guarantee robustness against novel incorrect logic that LLMs might produce. The claim, while technically true for their specific dataset, may imply a higher level of generalizability than is warranted.

3. The evaluation in Section 3 is convincing and insightful. Beyond showing that stronger models perform better, the paper offers little insight into why they succeed or fail.
- 3.1 The performance breakdown by algorithm category (Table 4) is a good start, but the analysis is shallow. For instance, why do models struggle with "Computational Geometry" or "Problems on Trees"?  What specific reasoning patterns are they failing to capture?
- 3.2 The paper could have included a qualitative analysis of model failures. Are the errors syntactic, logical, algorithmic (e.g., choosing a brute-force approach instead of dynamic programming), or implementation-related? This would provide far more value to researchers looking to improve these models.

**Questions:**

I wonder why the conclusion claimed by this paper is against the breaking news, like OpenAI and Google Gemini achieving golden medals in IOI competitions? What is the difference between AetherCode and the newest IOI contest?

https://www.reddit.com/r/MachineLearning/comments/1mnpqu7/n_openai_delivers_goldmedal_performance_at_the/

---

> ### Author Response · Authors · 2025-11-25
> **Rebuttal by Authors (1/3)**
>
> We thank the reviewer for your constructive comments and valuable time. Below, we address your concern point-by-point.
>
> ---
> ## Q1: The paper claims AetherCode is the first benchmark to systematically collect problems from premier programming competitions worldwide. This claim is overstated. Merging the data sources from several existing benchmarks and stating the comprehensiveness should not be a significant contribution to me.
>
> **While existing benchmarks have indeed incorporated some premier contests, their coverage is far from systematic.** The benchmarks mentioned by the reviewer suffer from two main limitations: (1) they are restricted to a very small number of specific contests; and (2) some rely on outdated data, posing a significant risk of data contamination. Specifically, we list the data sources for these benchmarks below:
>
> - ICPC-Eval: 11 ICPC contests (2023–2024)
> - USACO Bench: USACO problems (2011–2023)
> - OJBench: 4 ICPC contests + NOI problems (2016–2023)
> - LLM-Pros: 14 ICPC contests (2011–2024)
>
> AetherCode **systematically** collects the **latest** global **premier** programming contests from 2024–2025, offering a more diverse perspective for evaluating the coding and reasoning capabilities of LLMs. To the best of our knowledge, this is the first benchmark of its kind; therefore, we do not believe our claims are overstated.
>
> **AetherCode is not simply a merge of existing benchmarks.** We compared the data sources of AetherCode against 12 existing benchmarks, including LiveCodeBench, LiveCodeBench Pro, USACO Bench, ICPC-Eval, OJBench, LLM-Pros, CodeContests, APPS, CodeELO, TACO, HumanEval, and MBPP. Of the 78 contests covered by AetherCode, only 10 appear in existing benchmarks, while 68 are introduced for the first time. Consequently, the data sources of AetherCode significantly surpass previous works in terms of both comprehensiveness and novelty.
>
> *Contests appeared: EC-final-24, NAC-24, EC-Chengdu-24, EC-Nanjing-24, EC-Hangzhou-24, EC-HongKong-24, EC-Shanghai-24, EC-Shenyang-24, NWERC-24, CERC-24.*
>
>
> ---
> ## Q2: 100% TPR on our collected solution set is still not strong enough.
>
> **Do the solutions include subtle, hard-to-find bugs?** Yes. We conducted a case study on the solution set to confirm that it encompasses diverse error types, including the use of incorrect and inefficient algorithms, mishandling of corner cases, and memory limit exceeded (MLE) errors. However, manually reviewing every solution requires excessive labor; therefore, we cannot guarantee that every problem covers a fully comprehensive range of error types. During the solution collection phase, we can only maximize the quantity of solutions gathered.
>
> **A 100% TNR does not guarantee robustness against novel incorrect logic that LLMs might produce.** We also raised this exact concern in Section 2.3.3. We agree that relying solely on achieving a 100% TNR on this 30,000+ scale solution set is insufficient to achieve adequate coverage. Consequently, we recruited an elite team. Each member of this elite team holds at least three ICPC gold medals and has a minimum of two years of experience in competitive programming problem-setting. Specifically, this elite team further supplements missing corner cases and additionally writes various incorrect and inefficient solutions to verify the comprehensiveness of the test cases. We would like to respectfully point out that, **a set of test cases cannot cover all possible errors unless it exhausts all possible inputs.** Thus, test case completeness is never absolute, but rather a matter of relative optimization.
>
> **Our commitment to test case comprehensiveness significantly surpasses most prior studies.** Few existing works have utilized such a large-scale solution set to conduct a systematic analysis of test case quality. Indeed, some widely used benchmarks, such as LiveCodeBench, do not even incorporate procedures for verifying test case quality. Furthermore, to the best of our knowledge, this is the first code reasoning benchmark to engage a team of experts for the targeted optimization of test cases. Consequently, we are confident that the test cases in AetherCode are superior to those found in most previous works.

---

> ### Author Response · Authors · 2025-11-25
> **Rebuttal by Authors (2/3)**
>
> ## Q3: This paper offers little insight into why models succeed or fail.
>
> ### Analysis of failure reasons
>
> We have conducted a more in-depth analysis of the model's failure modes and incorporated it into the revised manuscript. We conducted our analysis in two aspects:
>
> **Quantitative Analysis.** We performed a statistical and quantitative analysis of failure reasons for all models based on their verdicts.
>
> **Qualitative Analysis.** We investigated the reasoning logic and implementation of o4-mini-high, specifically conducting a qualitative analysis of errors related to logic and implementation.
>
> The key findings from these analyses are summarized below.
>
> ## Key Findings
>
> | Model | Wrong Answer (%) | Time Limit (%) | Runtime Error (%) | Compile Error (%) |
> |-------|------------------|----------------|-------------------|-------------------|
> | o4-mini-high | 86.0 | 6.1 | 0.3 | 7.6 |
> | Gemini-2.5-Pro | 76.3 | 18.1 | 0.1 | 5.4 |
> | Seed-1.6-thinking-0715 | 79.1 | 15.2 | 0.1 | 5.6 |
> | DeepSeek-R1-0528 | 77.1 | 11.1 | 0.1 | 11.7 |
> | Qwen-3-235B-A22B-thinking | 81.3 | 12.3 | 0.0 | 6.4 |
> | Gemini-2.5-Flash | 79.7 | 11.4 | 0.1 | 8.9 |
> | GLM-4.5 | 71.0 | 10.5 | 0.0 | 18.5 |
> | Qwen-3-235B-A22B | 77.8 | 12.0 | 0.1 | 10.1 |
> | Qwen-3-32B | 77.7 | 13.8 | 0.1 | 8.5 |
> | Claude-Sonnet-4.5-thinking | 45.8 | 51.7 | 0.0 | 2.5 |
> | Claude-4-Opus-thinking | 48.2 | 48.3 | 0.0 | 3.5 |
> | Claude-4-Sonnet-thinking | 50.8 | 45.8 | 0.0 | 3.4 |
> | GPT-4.1 | 79.3 | 12.5 | 0.1 | 8.1 |
> | Qwen-3-8B | 69.2 | 9.1 | 0.1 | 21.7 |
> | Kimi-K2 | 77.0 | 7.2 | 0.0 | 15.7 |
> | DeepSeek-V3 | 82.8 | 9.2 | 0.1 | 7.8 |
> | Qwen-3-Coder-480B-A35B-Instruct | 78.9 | 15.3 | 0.1 | 5.8 |
> | Claude-4-Sonnet | 65.2 | 30.7 | 0.0 | 4.0 |
> | GPT-4o | 72.1 | 8.5 | 0.1 | 19.3 |
>
>
> In the quantitative analysis, we categorized the causes of error into four types based on the verdict: Wrong Answer (WA), Time Limit Exceeded (TLE), Runtime Error (RE), and Compile Error (CE). Our primary findings are summarized as follows:
>
> * **WA accounts for the largest proportion.** For most models, WA constitutes 70%–80% of all errors, followed by TLE.
> * **The Claude series exhibits a significantly higher TLE rate.** They have a tendency to prioritize generating logically correct solutions for difficult problems rather than optimizing for time complexity.
> * **CE rates vary significantly across different models.** The Claude series achieves the lowest CE rate, whereas some models, such as GLM4.5, show a very high CE rate.
> * **The primary cause of CE in GLM4.5 is the use of incorrect programming languages.** It retains a significant probability of providing a Python solution even when explicitly requested to use C++, indicating potential deficiencies in its instruction-following capabilities.
>
> In the qualitative analysis, we investigated the reasoning and code generation of o4-mini-high. The primary issues identified include the use of incorrect algorithmic logic, inefficient algorithms, failure to handle corner cases, and code implementation errors. Furthermore, we observed that o4-mini-high occasionally acknowledges its inability to solve a problem rather than providing an incorrect solution. We present the problems and cases in Appendix E and the supplementary material.
>
> We have incorporated these new results into the revised manuscript. For further details, please refer to Section 3.3, Appendix E, and the Supplementary Material (available for download on OpenReview).
>
> ### Why models particularly struggle with geometry and tree problems?
>
> The evaluated models generally underperformed on Tree and Geometry problems, a result we primarily attribute to the higher overall difficulty inherent in these categories. In Table 5, we present the difficulty distribution for each category; notably, the Tree category contains a particular high proportion of hard problems, while the Geometry category also exhibits slightly higher difficulty compared to others. This is likely the primary factor contributing to the lower scores. We conducted a small-scale examination of the failure patterns in these two categories, finding that they align with the common issues previously discussed. Specifically, failures in Tree problems were predominantly characterized by the use of incorrect or inefficient algorithms, whereas Geometry problems were more prone to errors involving corner cases and implementation. These findings are consistent with our expectations.

---

> ### Author Response · Authors · 2025-11-25
> **Rebuttal by Authors (3/3)**
>
> ## Q4: Why the conclusion claimed by this paper is against the breaking news, like OpenAI and Google Gemini achieving golden medals in IOI competitions? What is the difference between AetherCode and the newest IOI contest?
>
> This development occurred after the completion of our manuscript. However, even evaluating this event from a current perspective, we do not believe that OpenAI/Google achieving a gold medal at the IOI signifies that LLMs have fully 'conquered' competitive coding.
>
> **OpenAI models do demonstrate exceptional performance at IOI.** According to their technical report, o3 secured a gold medal without relying on hand-crafted, domain-specific strategies. **However, OpenAI wins IOI but struggle in many other contests.** We evaluated their reasoning model, o4-mini-high, and found it performed poorly in several instances. For example, in CCPC Final 2024, o4-mini-high solved only 1 out of 13 problems, ranking 118th out of 121 human teams. Across the entire AetherCode benchmark, o4-mini-high solved only one-third of the problems at Pass@1, and fewer than half at Pass@4. Admittedly, OpenAI and Google may possess more powerful unreleased models; however, to date, there is few evidence indicating that LLMs can consistently surpass human performance across the various coding contests.
>
> From this perspective, as LLM training datasets continue to expand, conducting true OOD evaluation becomes increasingly difficult. We try to fill this gap by build AetherCode, which aggregates a more diverse set of competitions, thereby assisting researchers in recognizing the genuine performance gaps that remain.
>
> **What is the difference between AetherCode and the newest IOI contest?** AetherCode is a collection of the latest premier coding contests around the world. The IOI is, of course, included in AetherCode; please refer to Table 7.

---

### Official Review · Reviewer_Lxkp · 2025-10-25

**Soundness:** 2
**Presentation:** 3
**Contribution:** 3
**Rating:** 6
**Confidence:** 4

**Summary:**

## Paper Summary

This paper presents AetherCode, a new benchmark designed to thoroughly evaluate the reasoning and programming capabilities of large language models (LLMs) using problems sourced exclusively from premier competitive programming contests such as IOI and ICPC. The authors identify two major deficiencies in existing benchmarks: (1) they lack sufficient difficulty and problem diversity, often consisting of relatively simple tasks (e.g., LeetCode-style problems), and (2) their evaluation suites frequently contain weak or incorrect test cases, resulting in inflated model performance. AetherCode addresses these issues by offering: (1) 456 carefully curated problems spanning multiple difficulty levels and algorithmic domains, (2) high-quality and rigorously validated test cases constructed through a Generator–Validator agent system augmented by expert audits, and (3) a comprehensive evaluation of 17 leading reasoning and non-reasoning LLMs, revealing a significant performance gap. The results clearly demonstrate that even top-tier LLMs can solve only a small fraction of the benchmark, confirming that substantial progress is still needed before LLMs can perform at the level of world-class human competitors.

**Strengths:**

## Strengths

1. Benchmark novelty and rigor. AetherCode is the first benchmark to systematically incorporate premier programming contest tasks, establishing a far more challenging and realistic evaluation setting.

2. Strong evidence of current reasoning limitations in LLMs. The steep decline in Pass@1 performance as problem difficulty increases highlights persistent weaknesses in deep algorithmic reasoning.

3. Clear and well-structured presentation. The paper is clearly written, logically organized, and easy to follow. The motivation, methodology, and findings are presented coherently, making the work accessible even for those not deeply familiar with competitive programming-style evaluation.

**Weaknesses:**

## Weaknesses

[1] Limited dataset scale. While the problems are high-quality, a total of 456 is still smaller than some widely adopted benchmarks, potentially limiting coverage in long-tail algorithmic domains.

[2] Clarification needed on contamination guarantees. Although timestamps are included, it remains unclear whether contamination checks confirm that evaluated LLMs were not trained on these problems prior to their incorporation into the dataset. Stronger contamination analysis or preventive measures would improve confidence in the benchmark’s integrity.

[3] Difficulty labeling procedure requires further explanation. The paper states that difficulty categories are based on human competition outcomes, but it would be helpful to describe the exact criteria or thresholds used, particularly for distinguishing “Hard” versus “Extreme.”

[4] Limited reporting on coverage of test cases. While the benchmark claims 100% TPR/TNR on a collected set of human solutions, metrics such as input space coverage or alignment with typical competitive programming corner-case patterns would further validate robustness.

[5] Missing related work. The paper would benefit from acknowledging and comparing against recent benchmarks that also focus on code LLMs.

1. Dynamic Benchmarking of Reasoning Capabilities in Code Large Language Models Under Data Contamination (ICML 2025)

2. DynaCode: A Dynamic Complexity-Aware Code Benchmark for Evaluating Large Language Models in Code Generation (ACL 2025 Finding)

3. Codereval: A benchmark of pragmatic code generation with generative pre-trained models (ICSE 2024)

4. EvoCodeBench: An Evolving Code Generation Benchmark Aligned with Real-World Code Repositories (Neurips 2025)

5. Evaluating and Improving LLM-based Competitive Program Generation

6. PPM: Automated Generation of Diverse Programming Problems for Benchmarking Code Generation Models (FSE 2024)

**Questions:**

1. How is the "difficult" determined in Table 1?

2. Do you have insights to scale the problem generation?

3. What is the coverage of the test cases?

4. Any ablation study of the effectiveness of the agent-based test case generation?

---

> ### Author Response · Authors · 2025-11-25
> **Rebuttal by Authors (1/2)**
>
> We thank the reviewer for your constructive comments and valuable time. Below, we address your concern point-by-point.
>
> ---
> ## Q1: A total of 456 is still smaller than some widely adopted benchmarks
> While it is true that AetherCode is not the largest code reasoning benchmark, its scale already surpasses that of several widely adopted benchmarks, such as HumanEval (164), CodeContests (165), and CodeELO (387). When observing other widely adopted reasoning benchmarks, some are also quite limited in size; for instance, AIME contains only 30 problems. Furthermore, although many benchmarks boast a large number of problems, they are often dated, which poses a risk of data contamination during evaluation. If we focus specifically on the volume of problems from the 2024–2025 period, AetherCode contains more problems than the majority of existing code reasoning benchmarks.
>
> ---
> ## Q2: Clarification needed on contamination guarantees.
>
> **We use strict cut-off dates to mitigate data contamination.** Following the approach of LiveBench and LiveCodeBench, we will provide an interactive leaderboard that allows users to specify an arbitrary cut-off date, filtering for models released prior to that date and problems after that date. Furthermore, AetherCode will be updated on a rolling basis to minimize future contamination.
>
> **Detecting data contamination is very difficult for code reasoning data.** While methods such as n-gram overlap analysis exist, they are often ineffective for code reasoning problems, which are usually trained using RL. Standard detection techniques struggle to capture contamination in RL data. Therefore, we believe that enforcing release cut-off dates is a more reliable strategy.
>
> ---
> ## Q3: Difficulty labeling procedure requires further explanation, particularly for distinguishing “Hard” versus “Extreme”.
> We rank problems within the same contest based on the number of participants who successfully solved them. For contests without leaderboards, as well as for determining the relative difficulty order across different contests, we rely on expert evaluation. Finally, based on the overall difficulty ranking of all problems, we divide the dataset into three roughly equal categories: Easy, Medium, and Hard. Problems that no human contestant was able to solve during a competition were specially classified as Extreme.
>
> ---
> ## Q4: Limited reporting on coverage of test cases. Metrics such as input space coverage or alignment with typical competitive programming corner-case patterns would further validate robustness.
>
> **Input space coverage.** The input space of typical competitive programming problems is extremely large, or even infinite, making it impossible to enumerate. Additionally, problems have different input structures, making it difficult to design a one-for-all coverage criterion.
>
> **Alignment with typical competitive programming corner-case patterns.** We recruited an elite team to further improve test case coverage beyond the 100% TNR bar. Each member of this elite team holds at least three ICPC gold medals and has a minimum of two years of experience in competitive programming problem-setting. Specifically, this elite team further supplements missing corner cases and additionally writes various incorrect and inefficient solutions to verify the comprehensiveness of the test cases. Therefore, we believe these test cases provide good coverage for various corner cases.
>
> ---
> ## Q5: Missing related work.
> We thank the reviewer for the suggested references. We have included them in the revised manuscript to make our literature review more comprehensive.
>
> ---
> ## Q6: Insights to scale the problem generation
> In this paper, we did not attempt to generate problems using models. Instead, we strictly collected expert-written problems. We acknowledge that using LLMs to generate problems is a promising direction for scaling up dataset size. However, we believe this approach still faces technical hurdles, particularly regarding the verification of the quality of LLM-generated problems. Furthermore, we consider such attempts more suitable for generating training data. For benchmarks, ensuring the accuracy of both the problems and the verifiers is more critical than scaling up the quantity.
>
> ---
> ## Q7: Any ablation study of the effectiveness of the agent-based test case generation?
> The Generator-Validator agent system was proposed in [1], which conducted ablation studies to verify its effectiveness. In our test case generation task, the G-V agent system achieved a TNR of 89.9% on its own, ultimately reaching 100% TNR with human intervention. We have added more details to Section 2.3.2.
>
> [1] CodeContests+: High-Quality Test Case Generation for Competitive Programming, EMNLP 2025 (Findings).

---

> ### Author Response · Authors · 2025-11-25
> **Rebuttal by Authors (2/2)**
>
> ## Q8: How is difficulty determined in Table 1?
> We categorize the difficulty of the datasets into three levels:
> 1. **Entry level (1 star)**: Involves basic algorithms and code implementation.
> 2. **Mixed level (2 stars)**: A dataset comprising a mixture of competition-level and entry-level problems.
> 3. **Competition level (3 stars)**: A dataset consisting exclusively of competition-level problems.

---

> > ### Comment · Reviewer_Lxkp · 2025-11-27
> > **response to authors rebuttal**
> >
> > [1] Regarding the dataset scale, while it is true that some existing benchmarks contain only a limited number of data instances, do you have any insights on how the dataset could be scaled in the future? Otherwise, it may become an endless cycle of proposing new benchmarks without fundamentally addressing the scalability issue. I acknowledge that scaling the dataset may be beyond the scope of this paper, but it would be very helpful if the authors could share some perspectives or potential directions for future scaling.
> >
> > [2] Could you provide more details about your cut-off implementation?
> >
> > [3] Could you offer additional clarification on the “difficult labeling” process? Since the labeling requires expert involvement, it would be valuable to understand how the authors ensure labeling consistency, manage disagreements among experts, and control the cost and scalability of expert-driven annotation.
> >
> > [4] Could you report code coverage metrics such as branch coverage or statement coverage?
> >
> > [5] The authors indicated that they will add the missing related work, so this concern has been addressed.
> >
> > [6] Same as [1].
> >
> > [7] The authors addressed my concern.
> >
> > [8] Same as [3].

---

> > > ### Author Response · Authors · 2025-12-03
> > > **Response by Authors (3/3)**
> > >
> > > ## Q4: Could you report code coverage metrics such as branch coverage or statement coverage?
> > >
> > > While **branch coverage** and **statement coverage** are standard white-box testing metrics in software engineering, we respectfully argue that they are not suitable for the context of competitive programming for the following reasons:
> > >
> > > 1. **Competitive programming is inherently black-box.** Competitive programming operates as a black-box testing scenario. For any given problem, there are multiple valid algorithmic approaches, and even a single algorithm can be implemented in numerous ways. Since the internal structure of candidate submissions varies significantly, it is impractical to predicate evaluation on the specific implementation details of a single "correct" code. Therefore, white-box metrics like branch and statement coverage are not applicable.
> > > 2. **Inability to measure test comprehensiveness.** Coverage metrics fail to reflect the comprehensiveness of a test suite in this domain. Even if we were to designate a specific "ground truth" implementation and achieve high coverage on it, this metric would not guarantee the quality of the test cases. For example, consider two input cases: one with $N=10$ and another with $N=10^6$. Both inputs might execute every line of a correct solution, resulting in 100% statement coverage for both. However, only the $N=10^6$ case is capable of rejecting an inefficient $O(N^2)$ solution. In conclusion, critical factors in competitive programming—such as time complexity constraints and the handling of edge cases in algorithm design—are effectively invisible to standard code coverage metrics.

---

> ### Author Response · Authors · 2025-12-03
> **Response by Authors (1/3)**
>
> We are pleased to learn that our previous response has addressed some of your concerns. We now address the remaining questions as follows:
>
> ---
> ## Q1: I acknowledge that scaling the dataset may be beyond the scope of this paper, but it would be very helpful if the authors could share some perspectives or potential directions for future scaling code reasoning datasets
>
> Competition-level programming problems are an extremely scarce resource. Creating these problems requires specialized domain experts, and top-tier global contests yield only a few hundred problems annually. Therefore, we agree that data synthesis is a promising research direction. We have summarized the current progress and challenges in code reasoning data synthesis, which primarily fall into two categories:
> 1. **Synthesizing test cases for real problem statements and ground truth solutions.** This approach is pragmatic because problem statements and solutions are relatively easy to access, whereas test cases are often not public. Related works, such as CodeContests+ [1] and HardTests [2], have shown that this method can achieve high alignment with human-crafted test cases. Crucially, this allows for the generation of problems that exceed the model's own capabilities; however, the volume is still constrained by the number of publicly available problems.
> 2. **Full-pipeline synthesis (Statement, Solution, and Test Cases).** The primary bottleneck here is the lack of reliable methods to verify the solvability of generated problems and the correctness of their solutions. Additionally, constrained by the model's own capabilities, this method cannot generate problems harder than the model itself. Research in this area is sparse. For instance, AutoCode [3] proposed a synthesis pipeline, but manual inspection revealed that only about 3.2% of the generated problems reached competition level. Furthermore, automatically identifying that specific 3.2% remains a significant challenge.
>
> While the works mentioned above focus largely on synthesizing **training** datasets, we believe that for **evaluation** sets, priority must be given to quality over quantity. Consequently, we incorporated automated test case generation methods to reduce manual effort, combined with rigorous review by a team of experts to ensure the highest possible data quality.
>
> ---
> ## Q2: Could you provide more details about your cut-off implementation?
>
> Data contamination is a critical challenge in LLM evaluation. Given that problems have distinct release dates (which correspond to contest times in AetherCode) and models possess specific knowledge cutoff dates, a prevailing solution is to implement a dynamic "live benchmark" [3,4] based on these temporal constraints.
>
> Specifically, AetherCode features a mechanism for rolling updates, where the leaderboard dynamically adjusts according to the selected time window. For instance, if the AetherCode time window is set to 202501–202505, the evaluation will strictly utilize problems released within this period. Crucially, only models with a knowledge cutoff date (or release date) prior to 202501 are included in this assessment. This strict temporal separation ensures that the evaluation data remains uncontaminated.
>
> **References**
>
> [1] CodeContests+: High-Quality Test Case Generation for Competitive Programming, EMNLP 2025 Findings
>
> [2] HardTests: Synthesizing High-Quality Test Cases for LLM Coding, arXiv 2505.24098
>
> [3] LiveBench: A Challenging, Contamination-Free LLM Benchmark, arXiv 2406.19314, https://livebench.ai/
>
> [4] LiveCodeBench: Holistic and Contamination Free Evaluation of Large Language Models for Code, ICLR 2025, https://livecodebench.github.io/leaderboard.html

---

> ### Author Response · Authors · 2025-12-03
> **Response by Authors (2/3)**
>
> ## Q3: Could you offer additional clarification on the “difficult labeling” process?
>
> We address the methodologies for **individual problem difficulty annotation** and **dataset difficulty classification** separately below, responding to the Reviewer’s previous Q3 and Q8, respectively.
>
> **I. Problem Difficulty Annotation (Response to Q3)**
>
> We employed the following three-step approach to annotate the difficulty of individual problems:
> 1. **Leveraging existing metadata.** During the data collection process, we retrieved leaderboards for the majority of the contests. For contests where a leaderboard was available, we established a relative difficulty ranking based on the number of successful submissions (solvers) or the average score per problem.
> 2. **Establishing a gold standard.** We selected five intermediate-level ICPC contests (~60 problems in total). Four domain experts collectively discussed and sorted these problems, dividing them evenly into three difficulty tiers: Easy, Medium, and Hard. This sorted list served as the "Gold Standard" anchor for subsequent annotations.
> 3. **Expert annotation.** The remaining ~400 problems were distributed among the four experts. Referring to the difficulty distribution in the Gold Standard set, experts categorized their assigned problems into the three tiers. For contests with leaderboards, experts only needed to determine the cutoff points (bars) between the tiers based on the pre-sorted order. For contests without leaderboards, experts evaluated and labeled each problem individually.
>
> This workflow allowed us to unify standards across experts while optimizing manual effort. We acknowledge that difficulty annotation inevitably involves some subjectivity, as problems cover diverse knowledge points and individual expertise varies. However, our labels reflect the human perspective on difficulty. This provides a valuable alternative perspective compared to recent works that classify difficulty based solely on model performance.
>
> **II. Dataset Difficulty Classification (Response to Q8)**
>
> Regarding the dataset difficulty classification presented in Table 1, our assessment primarily relies on the data source of these datasets (e.g., the originating website or platform). Our classification criteria are as follows:
>
> - **Entry level (1 star)**: Involves basic algorithms and code implementation.
> - **Mixed level (2 stars)**: A dataset comprising a mixture of competition-level and entry-level problems.
> - **Competition level (3 stars)**: A dataset consisting exclusively of competition-level problems.
>
> Justification for Specific Datasets:
> - **Entry Level:** HumanEval and MBPP are unequivocally classified as Entry level.
> - **Competition Level:** LiveCodeBench Pro, CodeELO, and USACO are fully competition-grade. LCB Pro and CodeELO are derived 100% from CodeForces, and USACO is itself a major algorithmic competition.
>
> Classifications for LiveCodeBench, APPS, and CodeContests may require further explanation:
>   - We consider LeetCode to be an Entry level platform. Since LiveCodeBench contains a significant volume of LeetCode problems, we classified it as Mixed level.
>   - Conversely, APPS and CodeContests are composed predominantly of competition-level problems, with only a small fraction sourced from platforms like LeetCode; therefore, they are classified as Competition level.
>
> These classifications were finalized through expert team discussions and also reference the difficulty grading found in prior literature [1, 5].
>
> **References**
>
> [1] CodeContests+: High-Quality Test Case Generation for Competitive Programming, EMNLP 2025 Findings
>
> [5] CodeElo: Benchmarking Competition-level Code Generation of LLMs with Human-comparable Elo Ratings, arXiv 2501.01257.

---

### Official Review · Reviewer_u8So · 2025-11-01

**Soundness:** 4
**Presentation:** 4
**Contribution:** 4
**Rating:** 10
**Confidence:** 3

**Summary:**

This paper introduces a coding benchmark based on top high-school and college programming contests (mainly IOI and ICPC). The problems are hand annotated for difficulty and categories of problems allowing a more finessed analysis. The test cases are tested against many contest submissions (both correct and incorrect) and additional test cases are generated by hand and automated systems and reviewed by experts. This level of programming contests is the natural next step for programming contest based benchmarks and based on what is stated in the paper a lot of care has gone into it's quality.

**Strengths:**

- The paper collects a substantial corpus of high-quality programming contest problems and formats them as a benchmark.
- The benchmark seems well annotated to allow for more detailed study.
- According to the paper much care was taken to ensure problem and especially test-case quality.

**Weaknesses:**

- The paper fails to detail where the example solutions came from and whether the creators of the problems or sample solutions consent to their work being used as an ML benchmark.
- It would be useful to know the difficulty breakdown in each category for assessing whether e.g. the models are really bad at Tree type problems or if the small number of tree problems all happen to be exceptionally difficult.

**Questions:**

- I think the IOI is aimed more at high-school students than middle-school L145

- "The performance of o4-mini-high and
Gemini-2.5-Pro is exceptional, establishing a significant performance gap that places them
in a tier of their own above other models. Furthermore, they are the only three models capable of
successfully solving problems at the “Extremely Difficult” level." you mention two models and then say they are the only *three* that solve extreme problems. Reword for clarity.

- If the authors have time, I would like to see the performance of Claude 4.5 Sonnet

- It would be nice to mention other efforts to improve benchmark quality and verify answers or test-cases. Two examples are:
PlatnumBench http://platinum-bench.csail.mit.edu/
SWE-bench Verified https://openai.com/index/introducing-swe-bench-verified/

**Details Of Ethics Concerns:**

How were the answers to the contest problems sourced? What is the copyright status of the original questions and are the creators ok with their use in ML research?

---

> ### Author Response · Authors · 2025-11-25
> **Rebuttal by Authors**
>
> We thank the reviewer for your constructive comments and acknowledgement of our contribution. Below, we address your concerns point-by-point.
>
> ## Q1: Copyright concerns about problems and solutions
>
> **Problems.** The problems included in AetherCode are derived from various competitions, a complete list of which is presented in Table 7. All contest materials utilized were obtained from publicly available sources on the internet. We conducted an audit of the licenses for these competitions. While some organizers provide explicit licensing terms—including IOI (CC BY), JOI (CC BY-SA 4.0), USACO (CC BY-NC-SA 4.0), and NOI (CC BY-NC 4.0). Official authorization or specific copyright details for other problems could not currently be verified. We have included a copyright statement (Appendix D) in the revised manuscript and remain committed to removing any potentially infringing problems upon the request of the copyright holders.
>
> **Solutions.** The solutions are not part of AetherCode and will not be publicly released. They are utilized exclusively during Test Case Construction to validate the quality of the test cases. One portion of these solutions is derived from public blogs and contestant submissions published by organizers. The remaining solutions were authored by our expert team.
>
> ---
> ## Q2: The difficulty breakdown in each category
>
> | Category | Easy % | Medium % | Hard % | Extreme % |
> |----------|--------|----------|--------|----------|
> | Basic Algorithms | 43.11% | 27.11% | 25.78% | 4.00% |
> | Common Techniques | 30.61% | 42.18% | 21.77% | 5.44% |
> | Computational Geometry | 16.67% | 30.56% | 47.22% | 5.56% |
> | Data Structures | 23.33% | 33.33% | 37.50% | 5.83% |
> | Dynamic Programming | 20.91% | 30.91% | 45.45% | 2.73% |
> | Graph Theory | 21.88% | 31.25% | 37.50% | 9.38% |
> | Mathematics | 25.00% | 32.29% | 36.46% | 6.25% |
> | Search | 28.00% | 32.00% | 36.00% | 4.00% |
> | Strings | 38.46% | 15.38% | 38.46% | 7.69% |
> | Tree Problems | 16.67% | 25.00% | 50.00% | 8.33% |
>
> Yes. We analyzed the difficulty distribution for each category. The proportion of difficult problems in the Tree category is indeed disproportionately high, which is likely the primary reason for the model's lower scores. We have revised the relevant discussion in Section 3.2 to emphasize the impact of this inconsistent difficulty distribution.
>
> ---
> ## Q3: Performance of Claude 4.5 Sonnet
>
> Claude Sonnet 4.5 Thinking achieved a Pass@1 score of 16.3. Please see revised Table 3 and Table 4 for the detailed scores.
>
> ---
> ## Q4: Mention other efforts to improve benchmark quality
> We thank the reviewer for the suggested references. We have included them in the revised manuscript to make our literature review more comprehensive.
>
> ---
> ## Q5: Other mistakes and typos
>
> We thank the reviewer for pointing out these mistakes. We have corrected them in the revised version.

---

### Official Review · Reviewer_dXGB · 2025-11-03

**Soundness:** 3
**Presentation:** 3
**Contribution:** 2
**Rating:** 4
**Confidence:** 4

**Summary:**

This paper introduces AetherCode, a benchmark designed to evaluate LLMs’ coding capabilities in premier programming competitions. The benchmark sources problems from top-tier contests such as IOI and ICPC, and constructs rigorous test cases through a hybrid approach combining an automated Generator–Validator (G-V) agent and professional human experts.

**Strengths:**

1. A harder and more discriminative coding benchmark is very timely and beneficial to the community, especially as existing datasets are approaching saturation for frontier reasoning models.
2. The inclusion of multiple ICPC/IOI medalists for test case construction is impressive and likely ensures very high evaluation quality. This is the main contribution of the paper from my opinion.

**Weaknesses:**

1. While the benchmark’s data source (premier contests) is appealing, the distinction from datasets such as LiveCodeBench Pro is not fully convincing. Those benchmarks already cover similar contest problems, so the contribution may appear incremental without a clearer articulation of what new research insights AetherCode enables.
2. The evaluation section mainly presents aggregate Pass@N scores but lacks qualitative or failure-case analyses. For instance, what types of reasoning or algorithmic errors are most common? The findings seem simply descriptive without such insights.

**Questions:**

1. The paper states that using the CodeForces judging service poses “compliance risks” since crawlers are explicitly prohibited by CodeForces. Could the authors provide an official citation or link to this policy?
2. The G-V Agent system is briefly described but remains under-specified. Could the authors elaborate on the exact mechanism? Also, how effective is the G-V agent alone, without expert correction?
3. How does AetherCode handle language diversity? Is it possible to extend the benchmark to multi-language evaluation?

---

> ### Author Response · Authors · 2025-11-25
> **Rebuttal by Authors (1/2)**
>
> We thank the reviewer for their constructive comments and valuable time. Below, we address each concern point-by-point.
>
> ---
>
> ## Q1: The distinction from datasets such as LiveCodeBench Pro is not fully convincing. Those benchmarks already cover similar contest problems.
>
> We would like to highlight the key distinctions regarding data sources between AetherCode and existing Code Reasoning Benchmarks:
>
> **First, AetherCode is exclusively derived from premier contests rather than online coding platforms.** In contrast, many existing benchmarks source problems from coding websites. For instance, benchmarks such as LiveCodeBench (from LeetCode.com, AtCoder.jp, CodeForces.com), LiveCodeBench Pro (from CodeForces.com), and CodeElo (from CodeForces.com) rely on online coding websites that have inherent limitations regarding problem design.
>
> Taking LiveCodeBench Pro—mentioned by the reviewer—as an example: as of this writing, LCB Pro consists solely of CodeForces problems. Standard CodeForces rounds are typically short-duration events (requiring an individual to solve ~6 problems in approximately 2 hours). Consequently, these problems often involve fewer reasoning steps per task and rarely require large-scale code implementations.
>
> In contrast, premier contests could offer significantly greater diversity and complexity. For example, Olympiad in Informatics (OI) contests often allow 5 hours to solve just 3 problems. This structure necessitates deeper reasoning and substantially larger code volumes. By aggregating problems from prestigious global contests (including various ICPC regionals and OI events), AetherCode achieves superior diversity compared to benchmarks restricted to online coding websites.
>
> (Note: While the authors of LCB Pro claim in their paper to include ICPC and OI problems, as of the time of this rebuttal, their dataset actually contains only CodeForces problems and lacks distinct collection of ICPC or OI problems.)
>
> **Second, while some prior works have attempted to incorporate premier contest problems, they are often limited to a narrow selection of events or rely on older data, which poses a high risk of data contamination.** Existing datasets include:
>
> * ICPC-Eval: 11 ICPC contests (2023–2024)
> * USACO Bench: USACO problems (2011–2023)
> * OJBench: 4 ICPC contests plus NOI problems (2016–2023)
> * LLM-Pros: 14 ICPC contests (2011–2024)
>
> AetherCode (v1) significantly outperforms these efforts in terms of breadth and currency. We have comprehensively collected problems from 78 of the latest premier global contests held between 2024 and 2025, providing a scale of coverage that far exceeds previous work.
>
> ---
> ## Q2: Lack qualitative or failure-case analyses
>
> We conducted a small-scale case study. The most common causes of model errors were the failure to find correct or sufficiently efficient algorithms, or the failure to handle corner cases. Less frequent error sources included compilation errors and instruction-following errors (e.g., failing to enclose the answer in a Markdown block). These errors stem from both model capabilities and the inherent nature of the problems; for instance, some problems possess a higher density of corner cases or stricter time complexity requirements. We consider these insights to be trivial and consistent with common sense. Since similar observations have been noted in prior literature [3,4], we did not claim these findings as a primary contribution of this work. We acknowledge that a deeper case analysis is valuable; however, it would require significant manual effort that exceeds our current resource constraints.
>
> ---
> ## Q3: Official citation or link to CodeForces crawling policy
>
> CodeForces outlines its crawling policy in https://codeforces.com/robots.txt. This file explicitly prohibits site-wide access for most crawlers and AI bots, limiting access strictly to the `/api` path for specific tools. Furthermore, the `/problemset/submit` interface explicitly forbids access by any user agent. Consequently, utilizing crawlers to automate code submission to CodeForces for LLM evaluation may pose compliance risks.

---

> ### Author Response · Authors · 2025-11-25
> **Rebuttal by Authors (2/2)**
>
> ## Q4: Could the authors elaborate on the exact test case generation mechanism? How effective is the G-V agent alone?
>
> We have incorporated the full test case generation workflow into Appendix C. For the concrete implementation of the G-V Agent, we refer to [1], which also includes ablation studies verifying its effectiveness. The G-V agent system independently achieved a TNR of 89.9% in our task, increasing to 100% with human intervention. Corresponding details have been added to Section 2.3.2.
>
> ---
> ## Q5: How does AetherCode handle language diversity? Is it possible to extend the benchmark to multi-language evaluation?
>
> AetherCode is not a language-specific benchmark. With the exception of very few problems that may rely on specific libraries or templates provided by the problem-setter, the vast majority of tasks support arbitrary programming languages. AetherCode will be compatible with SandboxFusion [2], enabling support for over 20 programming languages.
>
> **References**
>
> [1] CodeContests+: High-Quality Test Case Generation for Competitive Programming, EMNLP 2025 (Findings).
>
> [2] FullStack Bench: Evaluating LLMs as Full Stack Coders, arXiv 2412.00535
>
> [3] LiveCodeBench Pro: How Do Olympiad Medalists Judge LLMs in Competitive Programming? arXiv 2506.11928
>
> [4] Can Language Models Solve Olympiad Programming? arXiv 2404.10952.

---

> > ### Comment · Reviewer_dXGB · 2025-11-26
> >
> > Thank you for your response. Some of my questions have been addressed but I still hope to see a quantitative analysis of failure cases. I think beyond simply ranking models on a leaderboard, a good benchmark should also capture deeper insights that help researchers understand where improvements can be made. On a challenging benchmark like this, analyzing failure cases is especially critical. I appreciate the contributions of this work, but I am currently maintaining my score. I would raise it if statistical data on failure types (e.g., the percentage distribution of each failure category) were provided, even if only on a small scale.

---

> ### Author Response · Authors · 2025-11-27
> **Diagnosis of Failure Reasons**
>
> We sincerely thank the Reviewer for your time and for recognizing the contributions of our work. We appreciate the valuable suggestions provided, which have helped us significantly improve the quality of our manuscript. To specifically address your concerns, we have conducted a more in-depth failure case analysis, providing a quantitative breakdown of the error types and their underlying causes.
>
> We conducted our analysis in two aspects:
>
> **Quantitative Analysis.** We performed a statistical and quantitative analysis of failure reasons for all models based on their verdicts.
>
> **Qualitative Analysis.** We investigated the reasoning logic and implementation of o4-mini-high, specifically conducting a qualitative analysis of errors related to logic and implementation.
>
> The key findings from these analyses are summarized below.
>
> ## Key Findings
>
> | Model | Wrong Answer (%) | Time Limit (%) | Runtime Error (%) | Compile Error (%) |
> |-------|------------------|----------------|-------------------|-------------------|
> | o4-mini-high | 86.0 | 6.1 | 0.3 | 7.6 |
> | Gemini-2.5-Pro | 76.3 | 18.1 | 0.1 | 5.4 |
> | Seed-1.6-thinking-0715 | 79.1 | 15.2 | 0.1 | 5.6 |
> | DeepSeek-R1-0528 | 77.1 | 11.1 | 0.1 | 11.7 |
> | Qwen-3-235B-A22B-thinking | 81.3 | 12.3 | 0.0 | 6.4 |
> | Gemini-2.5-Flash | 79.7 | 11.4 | 0.1 | 8.9 |
> | GLM-4.5 | 71.0 | 10.5 | 0.0 | 18.5 |
> | Qwen-3-235B-A22B | 77.8 | 12.0 | 0.1 | 10.1 |
> | Qwen-3-32B | 77.7 | 13.8 | 0.1 | 8.5 |
> | Claude-Sonnet-4.5-thinking | 45.8 | 51.7 | 0.0 | 2.5 |
> | Claude-4-Opus-thinking | 48.2 | 48.3 | 0.0 | 3.5 |
> | Claude-4-Sonnet-thinking | 50.8 | 45.8 | 0.0 | 3.4 |
> | GPT-4.1 | 79.3 | 12.5 | 0.1 | 8.1 |
> | Qwen-3-8B | 69.2 | 9.1 | 0.1 | 21.7 |
> | Kimi-K2 | 77.0 | 7.2 | 0.0 | 15.7 |
> | DeepSeek-V3 | 82.8 | 9.2 | 0.1 | 7.8 |
> | Qwen-3-Coder-480B-A35B-Instruct | 78.9 | 15.3 | 0.1 | 5.8 |
> | Claude-4-Sonnet | 65.2 | 30.7 | 0.0 | 4.0 |
> | GPT-4o | 72.1 | 8.5 | 0.1 | 19.3 |
>
>
> In the **quantitative analysis**, we categorized the causes of error into four types based on the verdict: Wrong Answer (WA), Time Limit Exceeded (TLE), Runtime Error (RE), and Compile Error (CE). Our primary findings are summarized as follows:
>
> * **WA accounts for the largest proportion.** For most models, WA constitutes 70%–80% of all errors, followed by TLE.
> * **The Claude series exhibits a significantly higher TLE rate.** They have a tendency to prioritize generating logically correct solutions for difficult problems rather than optimizing for time complexity.
> * **CE rates vary significantly across different models.** The Claude series achieves the lowest CE rate, whereas some models, such as GLM4.5, show a very high CE rate.
> * **The primary cause of CE in GLM4.5 is the use of incorrect programming languages.** It retains a significant probability of providing a Python solution even when explicitly requested to use C++, indicating potential deficiencies in its instruction-following capabilities.
>
> In the **qualitative analysis**, we investigated the reasoning and code generation of o4-mini-high. The primary issues identified include the use of incorrect algorithmic logic, inefficient algorithms, failure to handle corner cases, and code implementation errors. Furthermore, we observed that o4-mini-high occasionally acknowledges its inability to solve a problem rather than providing an incorrect solution. We present the problems and cases in Appendix E and the supplementary material.
>
> We have incorporated these new results into the revised manuscript. For further details, please refer to Section 3.3, Appendix E, and the Supplementary Material (available for download on OpenReview).

---

> > ### Comment · Reviewer_dXGB · 2025-11-28
> >
> > Thanks for your reply and experiments provided. I appreciate your efforts made to improve the manuscript. Since the edit button is hidden currently, I would raise my score later.

---

### Comment · Area_Chair_hhVu · 2025-11-27
**Author Responses Are Ready - Please Review & Provide Feedback**

Dear Reviewers,

Thank you once again for your essential contributions to the review process. The authors have submitted their responses to your initial reviews.

I kindly ask you to carefully review the authors' responses for the papers you are handling. Your timely assessment of how the authors have addressed your original concerns is a critical step in reaching a final decision.

Please provide your feedback and any necessary updates to your reviews as soon as possible to ensure we can meet our tight schedule for the discussion phase.

Your prompt attention to this matter is highly appreciated.

Best regards,

Area Chair

---

### Meta-Review · Area_Chair_LXfB · 2026-01-07

**Summary:**

The paper introduces a benchmark aggregating recent competitive programming problems from all over the world. This is a nice effort and provides another evaluation resource for the community.
* The authors are encouraged to take the licensing issue seriously, and decide on what license they can release this under. While the few sources that they noted are released under CC-BY-(NC?), pointers and reference should be included as well. Probably fine on a best effort basis. Considerable effort went into setting these problems in the first place.
* Decide on a standard on metric that should be used on their benchmark (pass@1, pass@4?) ideally with higher k to partly make up for the small sample size
* release necessary meta data from their evaluation experiments for further analysis by the community

With ratings 2,4?,6,10, the paper is a bit controversial. The AC focused on if the main criticisms are addressed. For eg8w with rating of 2, all the main concerns are addressed well by the authors. On the concern about getting insights from failures, unfortunately this is easier asked than answered due to the nature of LLMs, and besides it is not necessary for the benchmark builder to understand all the nuances of the LLMs, so a reasonable effort should be sufficient here. Reviewer dXGB with a rating 4 stated they want to increase their score but was passed the time. Their concerns also seem to be mostly addressed.

**Reviewer Concerns:**

all addressed

**Reviewer Scores:**

one wanted to increase score

---

### Decision · Program_Chairs · 2026-01-26

Accept (Poster)